# Wasserstein Policy Optimization

**David Pfau** [1]   **Ian Davies** [1]   **Diana Borsa** [1]   **João G. M. Araújo** [1]   **Brendan Tracey** [1]   **Hado van Hasselt** [1]

## Abstract

We introduce Wasserstein Policy Optimization (WPO), an actor-critic algorithm for reinforcement learning in continuous action spaces. WPO can be derived as an approximation to Wasserstein gradient flow over the space of all policies projected into a finite-dimensional parameter space (e.g., the weights of a neural network), leading to a simple and completely general closed-form update. The resulting algorithm combines many properties of deterministic and classic policy gradient methods. Like deterministic policy gradients, it exploits knowledge of the *gradient* of the action-value function with respect to the action. Like classic policy gradients, it can be applied to stochastic policies with arbitrary distributions over actions – without using the reparameterization trick. We show results on the DeepMind Control Suite and a magnetic confinement fusion task which compare favorably with state-of-the-art continuous control methods.

## 1. Introduction

Reinforcement learning has made impressive progress in controlling complex domains with continuous actions such as robotics (Haarnoja et al., 2024), magnetic confinement fusion (Degrave et al., 2022) and game playing (Farebrother & Castro, 2024). A significant portion of this progress can be attributed to policy optimization methods, which directly optimize the parameters of a policy by stochastic gradient ascent on the expected long term return (e.g., Schulman et al., 2015a; 2017; Abdolmaleki et al., 2018). While unbiased estimates of the policy gradient can be computed directly from returns (Williams, 1992), many practical deep reinforcement learning algorithms for continuous control employ an actor-critic approach (Barto et al., 1983), which also estimates a value function to diminish the variance of

the policy update. The majority of these methods use a policy update derived from the classic policy gradient theorem for stochastic policies (Sutton et al., 1999), which applies to both discrete and continuous action spaces.

One notable exception is the class of algorithms derived from the *deterministic* policy gradient (DPG) theorem (Silver et al., 2014; Lillicrap, 2015). These use information about the *gradient* of the value in action space to update the policy, but are limited to deterministic policies, hindering exploration. Adding in stochasticity can be difficult to tune, and extensions that learn the variance (Heess et al., 2015; Haarnoja et al., 2018) rely on the reparameterization trick, which limits the class of policy distributions that can be used.

Here we present a new policy gradient method that uses gradients of the action value but can learn arbitrary stochastic policies. The method can be derived from the theory of Wasserstein gradient flows (Ambrosio et al., 2008), and projecting the nonparametric flow onto the space of parametric functions (e.g. neural networks) leads to a simple closed-form update that combines elements of stochastic and deterministic policy gradients. We give a high-level illustration of this in Fig. 1. The update is natural to implement as an actor-critic method. We call the resulting method *Wasserstein Policy Optimization* (WPO).

The paper is organized as follows. First, we review policy optimization for continuous control, and Wasserstein gradient flows. We then show how the latter can be used to derive WPO when applied to the former. We review prior work in this area and analyze the dynamics of WPO and how it relates to other policy gradients. We describe several extensions needed to convert WPO into a practical and competitive deep reinforcement learning algorithm. Finally, we compare its performance against several baseline methods on the DeepMind Control Suite (Tassa et al., 2018; Tunyasuvunakool et al., 2020) and a task controlling magnetic coils in a simulated magnetic confinement fusion device (Tracey et al., 2024). On a task where the task dimension can be varied arbitrarily, we find that WPO learns faster than baseline methods by an amount that increases as the task dimension grows, suggesting it may work well in very high dimensional action spaces. An open-source implementation

---

[1]Google DeepMind, London, UK. Correspondence to: David Pfau <pfau@google.com>.

*Proceedings of the $42^{nd}$ International Conference on Machine Learning*, Vancouver, Canada. PMLR 267, 2025. Copyright 2025 by the author(s).

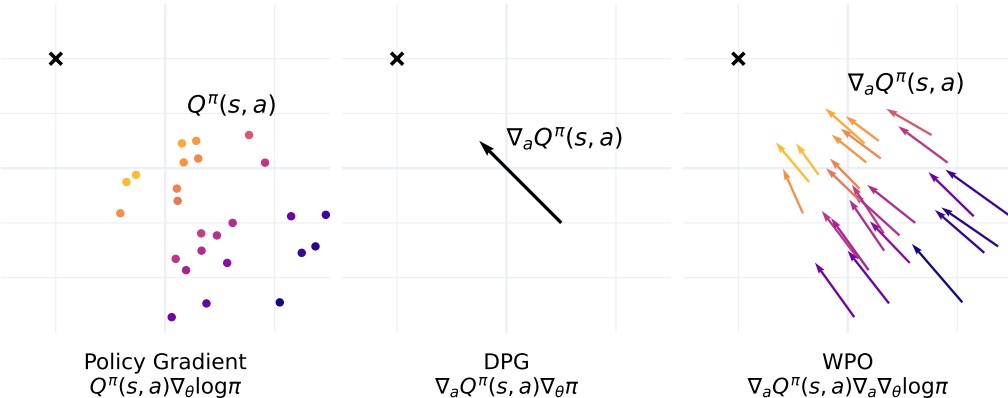

*Figure 1.* Conceptual illustration of how WPO combines elements of stochastic and deterministic policy gradient methods. Left: "classic" policy gradient. Samples are taken from a stochastic policy. Each sample contributes a scalar $Q^\pi(\mathbf{s}, \mathbf{a})$ factor to the gradient. Middle: deterministic policy gradient (DPG). A deterministic action is chosen and the policy gradient depends on the gradient of $Q^\pi(\mathbf{s}, \mathbf{a})$. Right: Wasserstein policy optimization (WPO). Samples are taken from a stochastic policy, as in classic policy gradient, but depend on the gradient of $Q^\pi$ with respect to the action, as in DPG.

of WPO is available in Acme (Hoffman et al., 2020)[1].

## 2. Method

### 2.1. Policy Gradient and Iteration Methods

We aim to find a policy $\pi(\mathbf{a}|\mathbf{s})$ which is a distribution over a continuous space of actions $\mathbf{a} \in \mathbb{R}^n$ conditioned on a state $\mathbf{s} \in \mathbb{R}^m$ that maximizes the expected long-term discounted return $\mathcal{J}[\pi] = \mathbb{E}_{\mathbf{a}_t \sim \pi, \mathbf{s}_t \sim \mathcal{P}}\left[\sum_{t=0}^T \gamma^t r_t\right]$ for a Markov decision process with transition distribution $\mathcal{P}(\mathbf{s}'|\mathbf{s}, \mathbf{a})$ and reward function $r(\mathbf{s}, \mathbf{a})$, where $r_t$ is shorthand for $r(\mathbf{s}_t, \mathbf{a}_t)$.

While "classic" policy gradient methods come in many forms (Schulman et al., 2015b), they are mostly variants of the basic update $\nabla_\theta \mathcal{J}[\pi_\theta] = \mathbb{E}_{\mathbf{a}_t \sim \pi, \mathbf{s}_t \sim \mathcal{P}}\left[\sum_{t=0}^T \Psi_t \nabla_\theta \log \pi_\theta(\mathbf{a}_t|\mathbf{s}_t)\right]$ and only differ in the choice of scalar $\Psi_t$. If $\Psi_t$ is entirely a function of the returns like $\sum_{t'=t}^T \gamma^{t'} r_{t'}$, then the policy can be optimized directly, as in REINFORCE (Williams, 1992). If $\Psi_t$ is the action-value function $Q^\pi(\mathbf{s}_t, \mathbf{a}_t)$ or some transformation of the action-value function like the advantage function, or softmax of the action-value, then both a policy and value function must be estimated simultaneously, which is standard in *actor-critic* methods (Barto et al., 1983). Additionally, it is common to add some form of trust region or regularization to prevent the policy update from changing too much on any step (Schulman et al., 2015a; 2017; Abdolmaleki

[1]https://github.com/google-deepmind/acme. Note that the implementation in Acme is *not* the version used for the experiments in this paper. However we have run this implementation on DeepMind Control Suite tasks and found qualitatively similar performance to that reported here.

et al., 2018). This is especially critical for achieving good performance with deep reinforcement learning.

One notable exception to this is deterministic policy gradients (DPG) (Silver et al., 2014; Lillicrap, 2015; Barth-Maron et al., 2018), which can be seen as the limit of the policy gradient update as the policy becomes deterministic. These algorithms were developed as early as the 1970s (Werbos, 1974) and were known under names such as 'action-dependent heuristic dynamic programming' (Prokhorov & Wunsch, 1997) or 'gradient ascent on the value' (van Hasselt & Wiering, 2007). As the name suggests, this only applies to deterministic policies $\pi$ that map state vectors to a single action vector, so $\mathbf{a}_t = \pi(\mathbf{s}_t)$. Then the deterministic policy gradient has the form $\nabla_\theta \mathcal{J}[\pi_\theta] = \mathbb{E}_{\mathbf{s}_t \sim \mathcal{P}}\left[\nabla_{\mathbf{a}_t} Q(\mathbf{s}_t, \mathbf{a}_t) \nabla_\theta \pi(\mathbf{s}_t)\right]$, where $\nabla_\theta \pi(\mathbf{s}_t)$ is the Jacobian of the deterministic policy. The appearance of the *gradient* of the value in action space in the policy gradient is potentially useful in high-dimensional action spaces. However, the restriction to deterministic policies makes exploration difficult. While there are extensions to DPG for learning a stochastic policy, such as SVG(0) (Heess et al., 2015) and SAC (Haarnoja et al., 2018), they rely on the reparameterization trick, which limits their generality.

Separately, policy iteration algorithms (Howard, 1960; Sutton & Barto, 2018) estimate the value of the current policy and then derive an improved policy based on these values. Iterating this converges to the optimal values and policy. These algorithms do not necessarily follow the gradient of the value. A modern example is MPO (Abdolmaleki et al., 2018), which updates its parametric policy (i.e., a neural network) towards the exponentiation of the current action values, using a target policy $\pi(a|s) \propto \exp(\frac{Q(\mathbf{s}, a)}{\tau})$

and minimizing a KL divergence with respect to this target.

## 2.2. Wasserstein Gradient Flows

It is possible to derive a true policy gradient for stochastic policies with a form similar to DPG, based on Wasserstein gradient flows. The theory of gradient flows is discussed in depth in (Ambrosio et al., 2008), and we review the relevant results here. Although the discussion here is fully general, we keep the notation as close to the RL notation as possible to avoid confusion.

Consider an arbitrary functional $\mathcal{J}[\pi]$ of a probability density $\pi(\mathbf{a})$, and let $\frac{\delta \mathcal{J}}{\delta \pi}$ be the functional derivative of $\mathcal{J}$. Then the following PDE will converge to a minimum of $\mathcal{J}$:

$$\frac{\partial \pi}{\partial t} = -\nabla_\mathbf{a} \cdot \left( \pi \left( -\nabla_\mathbf{a} \frac{\delta \mathcal{J}}{\delta \pi} \right) \right) \quad (1)$$

If we think of $-\nabla_\mathbf{a} \frac{\delta \mathcal{J}}{\delta \pi}$ as a velocity field, then this may be recognizable as the continuity equation from fluid mechanics, the drift term in the Fokker-Planck equation, or in a machine learning context as one way of writing the expression for the likelihood of a neural ODE (Chen et al., 2018). Problems in optimal transport can also be framed in terms of the continuity equation (Benamou & Brenier, 2000). The 2-Wasserstein distance, conventionally defined as

$$W_2^2(\pi_0, \pi_1) = \inf_{\rho \in \Gamma(\pi_0, \pi_1)} \int \rho(\mathbf{a}, \mathbf{b}) ||\mathbf{a} - \mathbf{b}||_2 d\mathbf{a} d\mathbf{b} \quad (2)$$

where $\Gamma(\pi_0, \pi_1)$ is the space of all joint distributions with marginals $\pi_0$ and $\pi_1$, can also be expressed as:

$$W_2^2(\pi_0, \pi_1) = \inf_{v_t \in V(\pi_0, \pi_1)} \int_0^1 \mathbb{E}_{\mathbf{a} \sim \pi_t} \left[ ||v_t(\mathbf{a})||_2 \right] dt \quad (3)$$

where $V(\pi_0, \pi_1)$ is the set of velocity fields $v_t$ such that, if $\pi_t = \pi_0$ at $t = 0$ and $\frac{\partial \pi_t}{\partial t} = -\nabla_\mathbf{a} \cdot (\pi_t (\nabla_\mathbf{a} v_t(\mathbf{a})))$, then $\pi_t = \pi_1$ at $t = 1$. From this dynamic formulation, it can be shown that the flow in Eq. 1 is the steepest descent on the functional $\mathcal{J}$ in the space of probability densities under the metric induced locally by the 2-Wasserstein distance.

In our case $\mathcal{J}[\pi]$ is the expected return in an MDP and we want to maximize this quantity. The functional derivative (wrt $\pi$) has the form:

$$\frac{\delta \mathcal{J}}{\delta \pi}(\mathbf{s}, \mathbf{a}) = \frac{1}{1 - \gamma} Q^\pi(\mathbf{s}, \mathbf{a}) d^\pi(\mathbf{s}) \quad (4)$$

where $d^\pi(\mathbf{s}) = (1 - \gamma) \sum_t \gamma^t \Pr(\mathbf{s}_t = \mathbf{s})$ is the discounted state occupancy function. A derivation is given in Sec. A.1 of the appendix. This is the functional generalization of the policy gradient in the tabular setting (see Agarwal et al. (2021), Eq. 7). The $d^\pi(\mathbf{s})$ term typically emerges implicitly in the update as the sampling frequency when interacting with the environment, and $(1 - \gamma)^{-1}$ is just a constant. In what follows, we focus on per-state updates where these terms do not appear.

## 2.3. Wasserstein Policy Optimization

To convert the theoretical results in the previous section into a practical algorithm, we need to approximate the PDE in Eq. 1 with an update to a parametric function such as a neural network. Starting at $\pi_\theta$ from the parametric family of functions, for a given infinitesimal $dt$ and flow $\frac{\partial \pi_\theta}{\partial t}$, we solve for the choice of infinitesimal $\Delta\theta$ which minimizes the KL divergence between the original distribution and the updated distribution. This gives the optimal direction to *cancel out* the flow, hence the minus sign in the KL below. It is well known that the KL divergence is approximated locally by a quadratic form with the Fisher information matrix (Pascanu & Bengio, 2013):

$$D_{\mathrm{KL}} \left[ \pi_\theta \middle|\middle| \pi_\theta + \frac{\partial \pi_\theta}{\partial t} dt - \nabla_\theta \pi_\theta \Delta\theta \right]$$

$$\approx \begin{pmatrix} dt \\ -\Delta\theta \end{pmatrix}^T \begin{pmatrix} \mathcal{F}_{tt} & \mathcal{F}_{t\theta}^T \\ \mathcal{F}_{t\theta} & \mathcal{F}_{\theta\theta} \end{pmatrix} \begin{pmatrix} dt \\ -\Delta\theta \end{pmatrix}$$

$$\mathcal{F}_{tt} = \mathbb{E}_\pi \left[ \frac{\partial \log \pi_\theta(\mathbf{a}|\mathbf{s})}{\partial t}^2 \right]$$

$$\mathcal{F}_{t\theta} = \mathbb{E}_\pi \left[ \frac{\partial \log \pi_\theta(\mathbf{a}|\mathbf{s})}{\partial t} \nabla_\theta \log \pi_\theta(\mathbf{a}|\mathbf{s}) \right]$$

$$= \int \frac{\partial \pi_\theta(\mathbf{a}|\mathbf{s})}{\partial t} \nabla_\theta \log \pi_\theta(\mathbf{a}|\mathbf{s}) d\mathbf{a}$$

$$\mathcal{F}_{\theta\theta} = \mathbb{E}_\pi \left[ \nabla_\theta \log \pi(\mathbf{a}|\mathbf{s}) \nabla_\theta \log \pi_\theta(\mathbf{a}|\mathbf{s})^T \right]$$

which is minimized at $\Delta\theta = \mathcal{F}_{\theta\theta}^{-1} \mathcal{F}_{t\theta}$.

Next, we derive a simple expression for $\mathcal{F}_{t\theta}$. We plug in $\frac{\partial \pi_\theta}{\partial t} = -\nabla_\mathbf{a} \cdot (\pi_\theta(\mathbf{a}|\mathbf{s}) \nabla_\mathbf{a} Q^\pi(\mathbf{a}, \mathbf{s}))$ for *ascent* on $\mathcal{J}[\pi]$:

$$\mathcal{F}_{t\theta} = \int \nabla_\theta \log \pi_\theta(\mathbf{a}|\mathbf{s}) \frac{\partial \pi_\theta(\mathbf{a}|\mathbf{s})}{\partial t} d\mathbf{a}$$

$$= -\int \nabla_\theta \log \pi_\theta(\mathbf{a}|\mathbf{s}) \nabla_\mathbf{a} \cdot (\pi_\theta(\mathbf{a}|\mathbf{s}) \nabla_\mathbf{a} Q^\pi(\mathbf{s}, \mathbf{a})) d\mathbf{a}$$

$$= \mathbb{E}_{\mathbf{a} \sim \pi} \left[ \nabla_\theta \nabla_\mathbf{a} \log \pi_\theta(\mathbf{a}|\mathbf{s}) \nabla_\mathbf{a} Q^\pi(\mathbf{s}, \mathbf{a}) \right] \quad (5)$$

This derivation is expanded in Sec. A.2 in the appendix, but mainly follows from integration by parts. This leads to a simple closed-form update for parametric policies which we call the *Wasserstein Policy Optimization update*:

$$\theta_{t+1} = \theta_t + \mathcal{F}_{\theta\theta}^{-1} \mathbb{E}_\pi \left[ \nabla_\theta \nabla_\mathbf{a} \log \pi(\mathbf{a}|\mathbf{s}) \nabla_\mathbf{a} Q^\pi(\mathbf{s}, \mathbf{a}) \right] \quad (6)$$

A similar parametric approximation was derived for finding the ground state energy of quantum systems (Neklyudov et al., 2023). While this application is quite different, the derivation follows from the same theory of gradient flows, arriving at a very similar expression. To the best of our

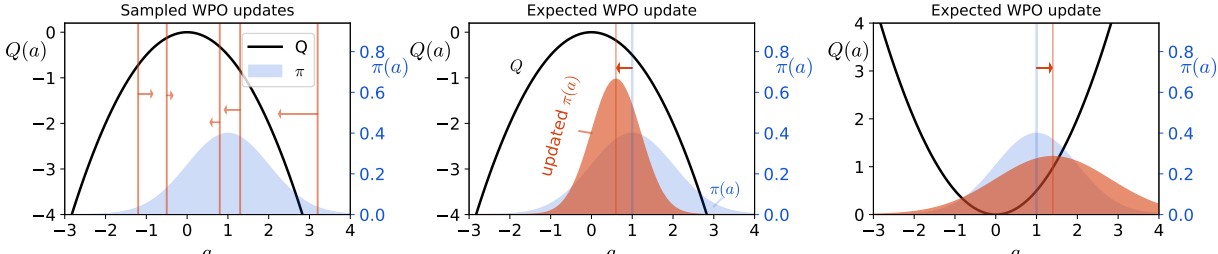

*Figure 2.* Concrete WPO updates for a single-variate normal policy for two different action-value functions. In the left and middle plots we consider $Q(a) = -a^2/2$, with an obvious optimum at $a = 0$, and a policy with $\mu = \sigma = 1$. The left plot shows the gradient on the mean at several sampled actions. These are averaged to produce the update, moving the mean towards the optimal action. The expected WPO update, as shown in the middle plot, is then $\Delta\mu = -\mu$ and $\Delta\sigma = -\sigma$. Both the mean and variance will decrease (or, more generally, move probability mass to the optimal action), as shown. Conversely, in the right plot we consider $Q(a) = a^2/2$ then $\Delta\mu = \mu$ and $\Delta\sigma = \sigma$, and both the mean and variance will increase, as shown. In all cases, the expected value of the resultant policy is increased.

knowledge we are the first to use this update for policy optimization in reinforcement learning. This somewhat idealized update requires the full Fisher information matrix and lacks some extensions needed to make policy optimization methods work in practice. We describe how to extend this update into a practical deep RL algorithm in Sec. 5.

## 3. Related Work

The general idea of using the Wasserstein metric in reinforcement learning has appeared in many forms. In Abdullah et al. (2019) the 2-Wasserstein distance is used to define Wasserstein Robust Reinforcement Learning, an algorithm that trains a policy which is robust to misspecification of the environment for which it is being trained. The Wasserstein distance appears here as a constraint on the transition dynamics of the environment, rather than as a way of defining learning dynamics. In Moskovitz et al. (2020), the Wasserstein metric is used locally as an alternative preconditioner in place of the Fisher information matrix in natural policy gradient (Kakade, 2001), while the conventional form of the policy gradient is still used. The Wasserstein distance has also been explored as an alternative to the KL divergence as a regularizer to prevent the policy from changing too quickly (Pacchiano et al., 2020).

The Wasserstein metric has also been used to define distances between *state* distributions, for various ends (Ferns et al., 2004; He et al., 2021; Castro et al., 2022), whereas we are focused on gradient flows in the space of *actions*.

In a combination of the above aims, (Metelli et al., 2019) and (Likmeta et al., 2023) use the Wasserstein distance to define a method to propagate uncertainty across state-action pairs in the Bellman Equation, with the aim of using that quantified uncertainty to better deal with the exploration-exploitation trade-off inherent to online reinforcement learning. In the context of actor-critic methods, this would

amount to a different way to update the critic, whereas WPO is a novel way to update the actor.

The idea of policy optimization as a Wasserstein gradient flow has appeared a few times in the literature. Richemond & Maginnis (2018) show that policy optimization with an entropy bonus can be written as a PDE similar to the expression in Sec. 2.2 with an additional diffusion term, but they stop short of deriving an update for parametric policies. The work of Zhang et al. (2018) also formulates policy optimization as a regularized Wasserstein gradient flow, but arrives at a more complicated DPG-like update that uses the reparameterization trick, similarly to SVG(0) and SAC. There is a close connection between Wasserstein gradient flows and Stein variational gradient descent (Chewi et al., 2020), but methods based on SVGD require expensive particle-based approximations (Messaoud et al., 2024).

## 4. Analysis

### 4.1. Gaussian Case

To better understand the mechanics of the WPO update, we analyze the simplified case where the policy is a single-variate normal distribution and the policy and value are not state-dependent. In this case we have:

$$\nabla_\mu \log \pi(a) = \frac{a - \mu}{\sigma^2} = -\nabla_a \log \pi(a)\,,$$

$$\nabla_\sigma \log \pi(a) = \frac{(a - \mu)^2}{\sigma^3} - \frac{1}{\sigma}\,,$$

$$\mathcal{F}_{\mu\mu} = \frac{1}{\sigma^2}\,, \mathcal{F}_{\sigma\sigma} = \frac{2}{\sigma^2}\,, \mathcal{F}_{\sigma\mu} = 0\,.$$

Let $\Delta_\mu \theta$ and $\Delta_\sigma \theta$ be the contributions to the update due to the gradients of the mean and variance, respectively, such

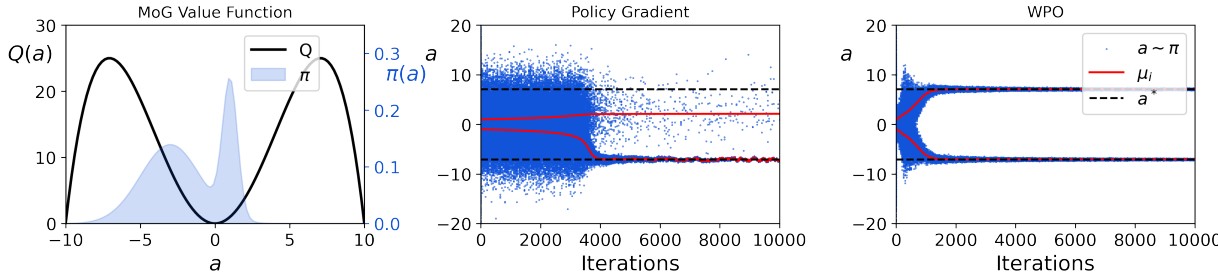

*Figure 3*. Concrete WPO learning for a one dimensional mixture of Gaussians policy for the non-concave action-value function $Q(a) = -\frac{1}{100}a^4 + a^2$. The left plot shows the action-value function and mixture of Gaussians policy. In the middle plot we show the evolution of the policy under a standard policy gradient update, both with samples from the policy and the change in the means of each mixture component. On the left we show the same evolution for WPO. WPO converges faster, is more stable around the optimum, and converges to both optima if the policy is initialized symmetrically.

that $\Delta\theta = \Delta_\mu\theta + \Delta_\sigma\theta$. For the mean, we then have

$$\Delta_\mu\theta = \mathcal{F}_{\mu\mu}^{-1}\mathbb{E}_\pi\left[\nabla_a Q(a)\nabla_a\nabla_\mu\log\pi(a)\nabla_\theta\mu\right]$$
$$= \sigma^2\mathbb{E}_\pi\left[\nabla_a Q(a)\nabla_a\frac{a-\mu}{\sigma^2}\nabla_\theta\mu\right]$$
$$= \mathbb{E}_\pi\left[\nabla_a Q(a)\nabla_\theta\mu\right] .$$

This is very similar to the DPG update: $\nabla_\mu Q(\mu)\nabla_\theta\mu$, except that we take the gradient of the action value at $a = \mu$, rather than sampling $a \sim \pi$. While this similarity holds for the normal distribution, it is not necessarily true in general.

For the variance we have:

$$\Delta_\sigma\theta = \mathcal{F}_{\sigma\sigma}^{-1}\mathbb{E}_\pi\left[\nabla_a Q(a)\nabla_a\nabla_\sigma\log\pi(a)\nabla_\theta\sigma\right]$$
$$= \frac{\sigma^2}{2}\mathbb{E}_\pi\left[\nabla_a Q(a)\nabla_\sigma\nabla_a\log\pi(a)\nabla_\theta\sigma\right]$$
$$= -\frac{\sigma^2}{2}\mathbb{E}_\pi\left[\nabla_a Q(a)\nabla_\sigma\frac{a-\mu}{\sigma^2}\nabla_\theta\sigma\right]$$
$$= \mathbb{E}_\pi\left[\frac{a-\mu}{\sigma}\nabla_a Q(a)\nabla_\theta\sigma\right]$$

This update is a little more complicated, but on inspection it also is quite intuitive. The variance increases when $(a - \mu)$ and $\nabla_a Q(a)$ have the same sign. So, when we sample an action, then we increase the variance if the gradient of $Q$ with respect to that action points even further away from the mean. If the gradient points back towards the mean, we will instead decrease the variance. Some special cases are illustrated in Figure 2.

Stochastic extensions of DPG such as SVG(0) (Heess et al., 2015) and soft actor-critic (SAC) (Haarnoja et al., 2018) use the reparameterization trick and define a gradient $\mathbb{E}_\eta[\nabla_a Q(s, a)\nabla_\theta\pi(s, \eta)]$, where $\pi$ denotes the action as deterministic function of the state, as well as a noise term $\eta$. For instance, we can define $\pi(s, \eta) = \mu(s) + \sigma(s) \circ \eta$, where $\eta \sim \mathcal{N}(0, I)$ and $\circ$ denotes the elementwise product. The mean and variance updates for the single-variate

normal distribution are then respectively $\mathbb{E}[\nabla_a Q(a)\nabla_\theta\mu]$ and $\mathbb{E}[\eta\nabla_a Q(a)\nabla_\theta\sigma]$. For this choice of parameterization, $\eta = (a - \mu)/\sigma$, which exactly coincides with WPO. SAC is similar to SVG(0), and extends this by including an entropy bonus to encourage exploration, as well as a mechanism based on double Q-learning (van Hasselt, 2010) to avoid value overestimations. Such extensions could be combined with WPO as well.

Note that the natural WPO update is independent of parameterization, while SVG(0) is not, so in general the two updates may still differ. For instance, suppose we have a task that requires non-negative actions, and we decide to use an exponential policy with components $a_i \sim \exp(a_i/\beta_i)/\beta_i$, where $\beta_i$ are learnable scale parameters for each action dimension. The Fisher for this distribution is diagonal with components $1/\beta_i^2$ on the diagonal. Then, the natural WPO update with respect to $\beta$ is

$$\mathcal{F}^{-1}\mathbb{E}_\pi\left[\nabla_a Q(a)\nabla_a\nabla_\beta\log\pi(a)\right]$$
$$= \text{diag}\left(\beta^2\right)\mathbb{E}_\pi\left[\nabla_a Q(a)\nabla_a\nabla_\beta(-\log\beta - a/\beta)\right]$$
$$= \text{diag}\left(\beta^2\right)\mathbb{E}_\pi\left[\nabla_a Q(a)\nabla_a(a/\beta^2 - 1/\beta)\right]$$
$$= \text{diag}\left(\beta^2\right)\mathbb{E}_\pi\left[\nabla_a Q(a)\text{diag}\left(1/\beta^2\right)\right]$$
$$= \mathbb{E}_\pi\left[\nabla_a Q(a)\right] .$$

We can reparameterize the policy for SVG(0) with a standard exponential $\eta$ with density $p(\eta = x) = \exp(-x)$ and then define $a = \beta \circ \eta$. Then, the SVG(0) update is

$$\mathbb{E}_\eta\left[\nabla_a Q(a)\nabla_\beta a\right] = \mathbb{E}_\eta\left[\nabla_a Q(a)\text{diag}\left(\eta\right)\right]$$
$$= \mathbb{E}_\pi\left[\nabla_a Q(a)\text{diag}\left(a/\beta\right)\right] ,$$

where we used $\eta = a/\beta$, by definition. This is not generally the same as the WPO update.

Not only does the natural WPO update coincide with DPG augmented with the reparameterization trick in *the Gaussian*

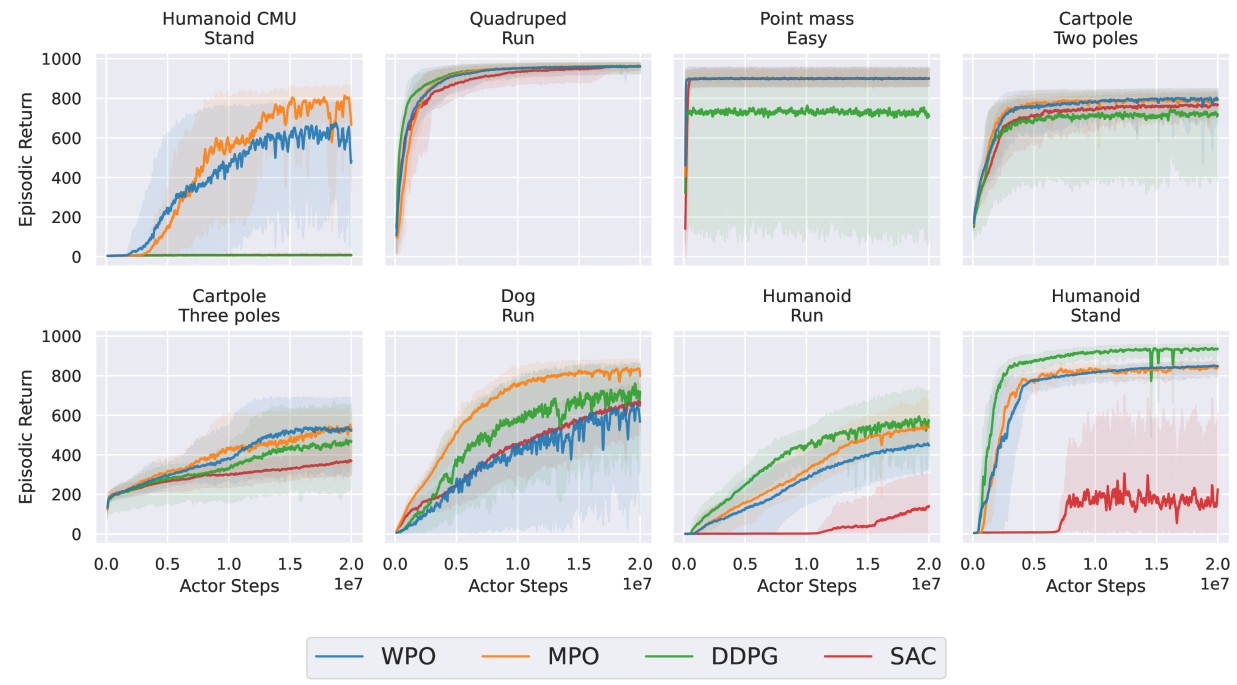

Figure 4. Results from selected DeepMind Control Suite tasks. Full results are in Fig. 7 in the appendix.

*case*, it also equals the standard policy gradient. Specifically,

$$\Delta_\mu \theta = \mathbb{E}_\pi \left[ \nabla_a Q(a) \nabla_\theta \mu \right] = \mathbb{E}_\pi \left[ \nabla_a Q(a) \right] \nabla_\theta \mu$$
$$= -\left[ \int_a Q(a) \nabla_a \pi(s,a) \right] \nabla_\theta \mu$$
$$\text{(as } \nabla_a \mathbb{E}_\pi[Q(a)] = 0)$$
$$= -\left[ \mathbb{E}_\pi [Q(a) \nabla_a \log \pi(a)] \right] \nabla_\theta \mu$$
$$= \mathbb{E}_\pi \left[ Q(a) \nabla_\mu \log \pi(a) \nabla_\theta \mu \right],$$
$$\text{(as } \nabla_a \log \pi = -\nabla_\mu \log \pi)$$

where the last line is just the expected policy gradient update for the parameters of the mean. A similar derivation, with the same result, can be done for the variance; this can be found in Appendix A.3. If we add those contributions together, we get the standard policy gradient $\mathbb{E}_\pi [Q(s,a)(\nabla_\mu \log \pi(s,a) \nabla_\theta \mu + \nabla_\sigma \log \pi(s,a) \nabla_\theta \sigma)] = \mathbb{E}_\pi [Q(s,a) \nabla_\theta \log \pi(s,a)]$.

These equivalences can be extended to multivariate normal distributions quite straightforwardly. This counterintuitive result suggests that there is essentially only one correct way to update a Gaussian policy, and that the major differences between approaches will only become clear when going beyond simple normal distributions over actions.

Importantly, even if the *expected* natural WPO update coincides with the *expected* policy gradient update, the *sampled* updates could still have dramatically different variance. For instance, consider an action-value function that is (locally) linear in the actions, such that $Q(\mathbf{s}, \mathbf{a}) = \mathbf{w}(\mathbf{s})^T \mathbf{a}$. Then

$\nabla_\mathbf{a} Q(\mathbf{s}, \mathbf{a}) = \mathbf{w}(\mathbf{s})$ does not depend on the action, and because the action-value gradient is then the same for each sampled action the WPO update for the mean of the policy will have zero variance. In contrast, in this case the standard policy gradient update will have non-zero variance. This observation is consistent with Fig. 1: the variance in the WPO updates will be low when locally all the gradients point roughly in the same direction.

## 4.2. Mixture of Gaussian Case

To understand the qualitative differences between different updates, we will have to move beyond the case of Gaussian policies. We consider the case of a one-dimensional mixture of Gaussians policy $\pi(a) = \sum_i \rho_i \mathcal{N}(a|\mu_i, \sigma_i)$ where $\sum_i \rho_i = 1$ are the mixture weights. Because the assignment of a sample to a mixture component is a discrete latent variable, the reparameterization trick cannot be used exactly (though it can be approximated through relaxations such as the concrete/Gumbel-softmax distribution (Maddison et al., 2017; Jang et al., 2017)) and so we exclusively consider WPO and classic policy gradient and not DPG/SVG(0). In this case, the dynamics of learning are too complex for closed-form results as in the previous section, so we consider an illustrative numerical example, with the action-value function $Q(a) = -\frac{1}{100}a^4 + a^2$, which has maxima at $\pm\sqrt{50}$. We initialize the policy with two components with $\rho_i = 0.5$, $\sigma_i = 10$ and $\mu_i = \pm 1$. We used a batch size of 1024 and learning rate of 0.003. For WPO, rather

than approximate the true Fisher information matrix, we rescale the gradients for $\rho_i$, $\mu_i$ and $\sigma_i$ by $\sigma_i^2$, a choice which is justified by the form of the FIM for a single Gaussian. We show results in Fig. 3. Despite being the same in expectation for the Gaussian case, policy gradient and WPO clearly have qualitatively different learning dynamics in the mixture-of-Gaussians case. WPO converges faster, is more stable around the minimum, and finds both local maxima (so long as the mixture components are initialized symmetrically). Notably, early in optimization, the variance of the policy actually *increases* for WPO, when the means of the mixture components are in the region with positive curvature, consistent with the intuition in Fig. 2.

## 5. Implementation

We make two modifications to the update in Eq. 6 to make WPO into a practical method. First, the use of the full natural gradient update is not practical for deep neural networks. It is tempting to simply drop it, but this could lead to serious numerical stability issues, as the $\nabla_{\mathbf{a}}\log\pi(\mathbf{a}|\mathbf{s})$ term in the update blows up as the policy converges to a deterministic result. Note that this is different from classic policy gradient methods, where natural gradient descent accelerates learning but is not strictly necessary (Kakade, 2001).

We could in theory use approximate second order methods such as KFAC (Martens & Grosse, 2015), but we have found that a much simpler approximation works well in practice. First, while the WPO update can be applied to arbitrary stochastic policies, we focus on normally-distributed policies $\pi_\theta(\mathbf{a}|\mathbf{s}) = \mathcal{N}(\mathbf{a}|\mu_\theta(\mathbf{s}), \Sigma_\theta(\mathbf{s}))$ where the covariance is constrained to be diagonal with elements $\sigma_i^2(\mathbf{s})$. The normal distribution with diagonal covariance has a diagonal Fisher information matrix with $\frac{1}{\sigma_i^2}$ for the mean element $\mu_i$ and $\frac{2}{\sigma_i^2}$ for the standard deviation $\sigma_i$.

Thus, rather than try to approximate the full Fisher information matrix, we may redefine the gradient of the log likelihood such that $\frac{\bar{\partial}}{\partial\mu_i}\log\mathcal{N}(\mathbf{a}|\mu,\Sigma) = \sigma_i^2\frac{\partial}{\partial\mu_i}\log\mathcal{N}(\mathbf{a}|\mu,\Sigma)$ and $\frac{\bar{\partial}}{\partial\sigma_i}\log\mathcal{N}(\mathbf{a}|\mu,\Sigma) = \frac{1}{2}\sigma_i^2\frac{\partial}{\partial\sigma_i}\log\mathcal{N}(\mathbf{a}|\mu,\Sigma)$. While this ignores the contribution of the gradients of $\mu$ and $\Sigma$ with respect to $\theta$ in the Fisher information matrix, it provides a qualitatively correct scaling that cancels out the tendency of the likelihood gradient to blow up as $\Sigma \to 0$. A similar heuristic was suggested in the original REINFORCE paper (Williams, 1992).

Secondly, we regularize the policy with a penalty on the KL divergence between the current and past policy. Regularization to prevent the policy from taking excessively large steps is standard practice in deep reinforcement learning (e.g., Schulman et al., 2015a; 2017). We similarly found that without regularization, WPO will prematurely collapse onto a deterministic solution and fail to learn, especially on

the fusion task from Tracey et al. (2024). While a variety of forms of KL regularization are used in continuous control, we closely follow the form used in MPO (Abdolmaleki et al., 2018). We can express this as a soft constraint, and modify the loss so that at each step we take a step towards solving

$$\max_\pi \mathbb{E}_{\mathbf{s}_t \sim \mathcal{P}}\left[\sum_{t=1}^{T}\gamma^t(\mathbb{E}_{\mathbf{a}_t \sim \pi}[r_t] - \alpha\mathrm{D}_{\mathrm{KL}}[\bar{\pi}(\cdot|\mathbf{s}_t)||\pi(\cdot|\mathbf{s}_t)])\right] \tag{7}$$

or express it as a hard constraint

$$\max_\pi \mathbb{E}_{\mathbf{a}_t \sim \pi, \mathbf{s}_t \sim \mathcal{P}}\left[\sum_{t=1}^{T}\gamma^t r_t\right]$$
$$s.t.\, \mathbb{E}_{\mathbf{s}_t \sim \mathcal{P}}\left[\mathrm{D}_{\mathrm{KL}}[\bar{\pi}(\cdot|\mathbf{s}_t)||\pi(\cdot|\mathbf{s}_t)] < \epsilon \tag{8}$$

which can be implemented by treating the $\alpha$ in the soft constraint as a Lagrange multiplier and performing a dual optimization step. Here $\bar{\pi}$ denotes a previous state of the policy, e.g., a target network. In either case, the gradient of the KL penalty is computed conventionally – only the gradient of the reward uses the approximate Wasserstein gradient flow and variance rescaling. While the KL divergence could be replaced by the Wasserstein distance, which would be more mathematically consistent and has been explored elsewhere as a regularizer (Richemond & Maginnis, 2018; Zhang et al., 2018; Pacchiano et al., 2020), we find the KL divergence works well in practice and leave it to future work to explore alternatives. We also note that while the KL penalty slows the convergence of WPO to a deterministic policy, it does not prevent it, as we show in Fig. 6.

## 6. Experiments

To evaluate the effectiveness of WPO, we evaluate it on the DeepMind Control Suite (Tassa et al., 2018; Tunyasuvunakool et al., 2020), a set of tasks in MuJoCo (Todorov et al., 2012). These tasks vary from one-dimensional actions, like swinging a pendulum, up to a 56-DoF humanoid. We additionally consider magnetic control of a tokamak plasma in simulation, a problem originally tackled by MPO in Degrave et al. (2022). On Control Suite, we compare WPO against both conceptually related and state-of-the-art algorithms which can be used in the same setting: Deep Deterministic Policy Gradient (DDPG; Lillicrap, 2015), Soft-Actor Critic (SAC; Haarnoja et al., 2018), and Maximum a Posteriori Policy Optimization (MPO; Abdolmaleki et al., 2018).

Our training setup is similar to other distributed RL systems (Hoffman et al., 2020): we run 4 actors in parallel to generate training data for the Control Suite tasks, and 1000 actors for the tokamak task. For WPO, the policy update uses sequences of states from the replay buffer, which may come from an old policy, but the actions are resampled from the current policy, making the algorithm effectively off-policy

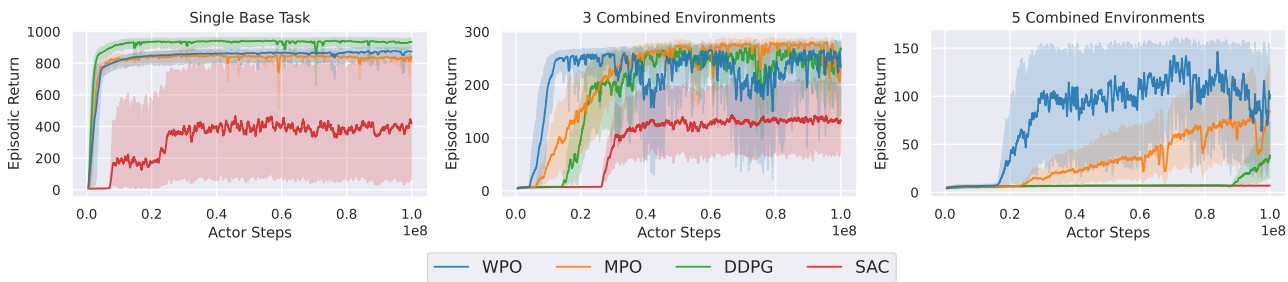

*Figure 5.* Plots of reward from various agents on combined Humanoid Stand environments. Left to right: 1, 3 and 5 replicated environments (21, 65 and 105 action dimension). Solid line denotes the mean and the shaded region highlights the minimum and maximum over 5 seeds. As the number of replicas grows, WPO is able to learn faster than other methods by a larger margin.

for states but on-policy for actions. MPO is implemented similarly. We used the soft KL penalty in Eq. 7 for Control Suite tasks, as we found the hard KL penalty did not noticeably improve results, but used the hard KL penalty for the fusion task for consistency with previously published results. Separate KL penalties with different weights were put on the mean and variance of the policy, as described in Song et al. (2019). We found that the penalty on the mean had little effect on stability and mainly slowed convergence, while the penalty on the variance noticeably helped stability. Training hyperparameters are listed in Sec. B and an outline of the full training loop is given in Alg. 1 in the appendix. For each RL algorithm, the same hyperparameters were used for each control suite environment.

For the critic update we use a standard $n$-step TD update with a target network:

$$\delta_{TD} = \left[ \sum_{\tau=0}^{n} r_{t+\tau} + \gamma^n \bar{V}(\mathbf{s}_{t+n}) \right] - Q_\theta(\mathbf{s}_t, \mathbf{a}_t) \quad (9)$$

where the target value $\bar{V}(\mathbf{s}_{t+n})$ is approximated by sampling multiple actions from $\pi$. This target value could be the mean of the target action value network, the maximum, or something in between. In MPO, the softmax over samples is theoretically optimal, so we use that for Control Suite. We use the maximum for WPO on all Control Suite tasks, as that worked well on the hardest domains, but use the mean for both MPO and WPO on fusion tasks, as the performance is significantly better.

### 6.1. DeepMind Control Suite

Figure 4 shows results from a subset of the DeepMind Control Suite. We selected tasks which show the full range of WPO's performance from excelling to struggling. Learning curves for all Control Suite tasks are in Fig. 7 in the appendix. While no single algorithm uniformly outperformed all others on all tasks, it is notable that DDPG and SAC converged to lower rewards and occasionally struggled to take

off on a number of tasks. This is even noticed for relatively low dimensional tasks (particularly with sparse reward). WPO robustly takes off and is in the same range as the best performing method across nearly all tasks. Through hyperparameter tuning, we noticed that SAC is particularly sensitive to the weighting of its entropy objective. This meant that finding generally hyperparameters across all tasks led to difficult trade-offs of stability and performance. We note that WPO and MPO demonstrated greater out-of-the-box generalisation across Control Suite. On the Humanoid CMU domain, one of the highest dimensional tasks in the Control Suite, neither SAC nor DDPG took off at all, while WPO made progress on all tasks and, in the case of the Walk task, initially learned faster than MPO. On Dog, another high-dimensional domain, WPO converged to roughly the same final reward, but often took longer. Dog is the only domain where the state space is larger than the observation space, suggesting that WPO may have difficulty in partially observed settings. We experimented with several choices of nonlinearity, squashing function (see Sec. C.3 and Fig. 12), KL regularization weight, and critic bootstrapping update for WPO. This is still less than the years of experimentation which has gone into many other popular continuous control algorithms (Huang et al., 2022). This shows that, while DDPG or SAC may struggle to learn on certain tasks, WPO almost always makes some learning progress, and is often comparable to state-of-the-art methods for these tasks, even without significant tuning.

### 6.2. Combined Tasks

Methods that use action-value gradients may perform well in high-dimensional action spaces, but no Control Suite task goes beyond a few dozen dimensions. To evaluate the effectiveness of WPO in higher dimensional action spaces, we construct tasks that consist of controlling many replicas of a Control Suite environment simultaneously with a single centralized agent. Specifically, the action and observation vectors are concatenated, and rewards are combined with a

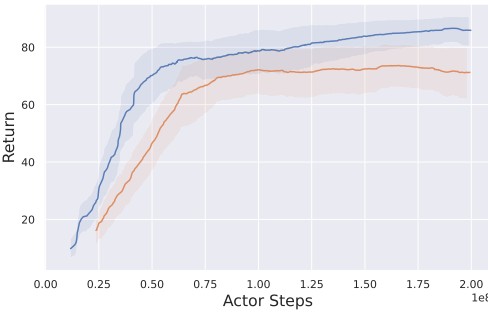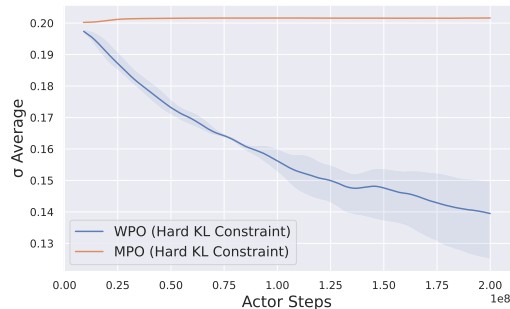

*Figure 6.* Reward and policy average standard deviation evolution throughout training on the fusion task discussed in Section 6.2

`SmoothMin` operator (see Eq. 31 in the Appendix), biasing the reward towards lower performing replicas to encourage learning across all replicas. We use the same agent hyperparameters as we used in the standard control suite benchmark no matter the number of replicas. We selected Humanoid - Stand as the base task due to its relatively high dimensional action space (21) and moderate difficultly.

The results in figure 5 demonstrate that as the number of replicas grows, WPO continues to be able to learn across the environments. In the singleton case, all methods are qualitatively similar except SAC which converges to a lower reward than other learning algorithms. As the number of replicas grows, WPO takes off earlier in training than MPO, which takes off earlier than DDPG, with SAC being the slowest to take off even in the singleton case. This trend becomes more pronounced for larger numbers of replicas, even if the asymptotic performance is similar for all methods. This suggests that for tasks with hundreds of action dimensions, WPO may be able to learn faster than other methods.

### 6.3. Fusion

Degrave et al. (2022) used MPO to discover policies to control the magnetic coils of the TCV tokamak, a toroidal fusion experiment (Duval et al., 2024). The resulting policies successfully ran at 10kHz to hold the plasma stable. Follow-on work (Tracey et al., 2024) explored methods to improve the accuracy of RL-derived magnetic control policies. We run WPO on a variant of the `shape_70166` task considered therein, modified to reward higher control accuracy (details in Sec. B.3). This task has 93 continuous measurements, a 19-dimensional continuous action space and lasts 10,000 environment steps, simulating a 1s experiment on TCV.

We compare the performance of WPO (with cube-root squashing as described in Sec. C.3) alongside MPO in Fig. 6. Both MPO and WPO used the default network and settings as in Tracey et al. (2024), including the hard KL regularization as in Eq. 8. WPO achieved a slightly higher reward than

MPO. Interestingly, we see a notable difference in the adaptation of the policy variance between the two algorithms. WPO evolves in the direction of a deterministic policy as training evolves, as one would expect for a fully observed environment, while MPO maintains approximately constant policy variance on average. We see similar results in B.3 on the original `shape_70166` task, where MPO and WPO achieve similar rewards with different policy adaptation behavior. These results show WPO as a viable alternative to MPO in this complex real-world task.

## 7. Discussion

We derived a novel policy gradient method from the theory of optimal transport: Wasserstein Policy Optimization. The resulting gradient has an elegantly simple form, closely resembling DPG, but can be applied to learn policies with arbitrary distributions over actions. On the DeepMind Control Suite, WPO is quite robust, performing comparably to state-of-the-art methods on most tasks, despite little tuning. Results on magnetic confinement fusion domains show that WPO can be applied to diverse tasks beyond simulated robotics. Promising initial results suggest it can learn quickly on tasks with over 100 action dimensions. We hope these results can help unlock performant algorithms for very larger control problems, and inspire researchers to develop new, challenging high-dimensional benchmarks in continuous control with hundreds of action dimensions or more.

While we chose a particular instantiation of WPO for the experiments, the WPO update is general and can be the foundation for many different implementations. We have only begun to scratch the surface of how WPO can be applied, and we are hopeful that extensions such as non-Gaussian policies or more advanced critic updates (Bellemare et al., 2017) could fully exploit the advantages of this new method. This could greatly increase the performance and scope of problems to which WPO can be applied.

## Acknowledgements

We would like to thank Maria Bauza, Nimrod Gileadi, Abbas Abdolmaleki for assistance running experiments, and Tom Erez, Jonas Buchli, Alireza Makhzani, Guy Lever, Leonard Hasenclever and Kirill Neklyudov for helpful discussions.

## Impact Statement

The present work can be useful for any reinforcement learning domain with continuous actions, such as robotics or industrial control. As this is a fundamental algorithmic advance, it does not favor any particular socially beneficial or dangerous application, but instead could enable progress across all applications. Other policy optimization methods like PPO are widely used in alignment of generative AI models, so WPO could conceivably have applications in alignment in domains with continuous action spaces, which could help to avert undesirable behavior.

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

# A. Extended Derivations

## A.1. Functional Derivative of the Loss

To prove that WPO is a true policy *gradient* method, and not simply a policy *improvement* method, we need to prove that the functional derivative of the policy return with respect to the policy has the form in Eq. 4. While this is a known result in the tabular case (see Agarwal et al. (2021), Eq. 7), we give a derivation in the case of continuous action and state spaces for completeness here. Suppose we perturb a policy $\pi(\mathbf{a}|\mathbf{s})$ by an amount $\delta\pi(\mathbf{a}|\mathbf{s})dt$, such that $\int d\mathbf{a}\delta\pi(\mathbf{a}|\mathbf{s}) = 0 \,\forall\, \mathbf{s}$. Then in the $dt \to 0$ limit the change to the expected long-term reward is

$$\lim_{dt \to 0} \frac{\mathcal{J}[\pi + \delta\pi dt]}{dt} - \mathcal{J}[\pi] = \delta\mathcal{J}[\pi] = \int d\mathbf{a}d\mathbf{s}\frac{\delta\mathcal{J}[\pi]}{\delta\pi}(\mathbf{s}, \mathbf{a})\delta\pi(\mathbf{s}|\mathbf{a})$$

to solve for $\frac{\delta\mathcal{J}[\pi]}{\delta\pi}$ we plug in this perturbation into the definition of the objective function:

$$\mathcal{J}[\pi] = \mathbb{E}_\tau\left[\sum_{t=0}^{T} \gamma^t r_t\right] \tag{10}$$

$$= \sum_{t=0}^{T} \gamma^t \int d\tau_{0:t} r_t \mathcal{P}(\mathbf{s}_0) \prod_{t'=0}^{t} \pi(\mathbf{a}_{t'}|\mathbf{s}_{t'}) \prod_{t'=0}^{t-1} \mathcal{P}(\mathbf{s}_{t'+1}|\mathbf{a}_{t'}, \mathbf{s}_{t'}) \tag{11}$$

$$= \sum_{t=0}^{T} \gamma^t \int d\tau_{0:t} r_t p^\pi(\tau_{0:t}) \tag{12}$$

where $\tau$ is shorthand for full-length trajectories $\mathbf{s}_0, \mathbf{a}_0, \ldots, \mathbf{s}_T, \mathbf{a}_T$, $\tau_{0:t}$ is shorthand for truncated trajectories $\mathbf{s}_0, \mathbf{a}_0, \ldots, \mathbf{s}_t, \mathbf{a}_t$ and $p^\pi(\tau_{0:t})$ denotes the probability of a trajectory, combining environment transitions and policy steps. Plugging in $\pi \to \pi + \delta\pi dt$ into Eq 11 and expanding, the perturbed value is then:

$$\delta\mathcal{J}[\pi] = \lim_{dt \to 0} \frac{\mathcal{J}[\pi + \delta\pi dt]}{dt} - \mathcal{J}[\pi] \tag{13}$$

$$= \sum_{t=0}^{T} \gamma^t \int d\tau_{0:t} r_t \mathcal{P}(\mathbf{s}_0) \prod_{t'=0}^{t} \pi(\mathbf{a}_{t'}|\mathbf{s}_{t'}) \prod_{t'=0}^{t-1} \mathcal{P}(\mathbf{s}_{t'+1}|\mathbf{a}_{t'}, \mathbf{s}_{t'}) \sum_{t'=0}^{t} \frac{\delta\pi(\mathbf{a}_{t'}|\mathbf{s}_{t'})}{\pi(\mathbf{a}_{t'}|\mathbf{s}_{t'})} \tag{14}$$

$$= \sum_{t=0}^{T} \gamma^t \int d\tau_{0:t} r_t p^\pi(\tau_{0:t}) \sum_{t'=0}^{t} \frac{\delta\pi(\mathbf{a}_{t'}|\mathbf{s}_{t'})}{\pi(\mathbf{a}_{t'}|\mathbf{s}_{t'})} \tag{15}$$

$$= \sum_{t=0}^{T} \gamma^t \int d\tau_{0:t} r_t p^\pi(\tau_{0:t}) \sum_{t'=0}^{t} \frac{\delta\pi(\mathbf{a}_{t'}|\mathbf{s}_{t'})}{\pi(\mathbf{a}_{t'}|\mathbf{s}_{t'})} \int d\tau_{t+1:T} p^\pi(\tau_{t+1:T}|\tau_{0:t}) \tag{16}$$

$$= \sum_{t=0}^{T} \gamma^t \int d\tau r_t p^\pi(\tau) \sum_{t'=0}^{t} \frac{\delta\pi(\mathbf{a}_{t'}|\mathbf{s}_{t'})}{\pi(\mathbf{a}_{t'}|\mathbf{s}_{t'})} \tag{17}$$

$$= \sum_{t'=0}^{T} \int d\tau p^\pi(\tau) \frac{\delta\pi(\mathbf{a}_{t'}|\mathbf{s}_{t'})}{\pi(\mathbf{a}_{t'}|\mathbf{s}_{t'})} \sum_{t=t'}^{T} \gamma^t r_t \tag{18}$$

$$= \sum_{t'=0}^{T} \int d\tau_{0:t'} \mathcal{P}(\mathbf{s}_0) \prod_{t''=0}^{t'-1} \pi(\mathbf{a}_{t''}|\mathbf{s}_{t''}) \mathcal{P}(\mathbf{s}_{t''+1}|\mathbf{a}_{t''}, \mathbf{s}_{t''}) \delta\pi(\mathbf{a}_{t'}|\mathbf{s}_{t'}) \gamma^{t'} Q^\pi(\mathbf{s}_{t'}, \mathbf{a}_{t'}) \tag{19}$$

$$= \int d\mathbf{s}d\mathbf{a}\delta\pi(\mathbf{a}|\mathbf{s}) Q^\pi(\mathbf{s}, \mathbf{a}) \sum_{t'=0}^{T} \gamma^{t'} \mathrm{Pr}^\pi(\mathbf{s}_t = \mathbf{s}) \tag{20}$$

$$\frac{\delta\mathcal{J}[\pi]}{\delta\pi} = Q^\pi(\mathbf{s}, \mathbf{a}) \sum_{t=0}^{T} \gamma^t \mathrm{Pr}^\pi(\mathbf{s}_t = \mathbf{s}) = \frac{1}{1-\gamma} Q^\pi(\mathbf{s}, \mathbf{a}) d^\pi(\mathbf{s}) \tag{21}$$

where $\Pr^\pi(\mathbf{s}_t = \mathbf{s})$ is the marginal probability of the state $\mathbf{s}$ at time $t$ under the policy $\pi$ and $d^\pi = (1-\gamma)\sum_t \gamma^t \Pr^\pi(\mathbf{s}_t = \mathbf{s})$ is the discounted state occupancy function. Thus the action-value function $Q^\pi$ is *almost* the functional derivative $\frac{\delta \mathcal{J}[\pi]}{\delta \pi}$, but with an additional discounted occupancy over the states.

## A.2. Projection Onto a Parametric Function Space

Here we give a more complete version of the derivation in Sec. 2.3

$$\mathcal{F}_{t\theta} = \int \nabla_\theta \log \pi_\theta(\mathbf{a}|\mathbf{s}) \frac{\partial \pi_\theta(\mathbf{a}|\mathbf{s})}{\partial t} d\mathbf{a} \tag{22}$$

$$= -\int \nabla_\theta \log \pi_\theta(\mathbf{a}|\mathbf{s}) \nabla_\mathbf{a} \cdot (\pi_\theta(\mathbf{a}|\mathbf{s}) \nabla_\mathbf{a} Q^\pi(\mathbf{s},\mathbf{a})) \, d\mathbf{a} \tag{23}$$

$$= -\int \nabla_\theta \log \pi_\theta(\mathbf{a}|\mathbf{s}) \left[ \pi_\theta(\mathbf{a}|\mathbf{s}) \nabla_\mathbf{a}^2 Q^\pi(\mathbf{s},\mathbf{a}) + \nabla_\mathbf{a} \pi_\theta(\mathbf{a}|\mathbf{s}) \nabla_\mathbf{a} Q^\pi(\mathbf{s},\mathbf{a}) \right] d\mathbf{a} \tag{24}$$

$$= -\int \nabla_\theta \pi_\theta(\mathbf{a}|\mathbf{s}) \nabla_\mathbf{a}^2 Q^\pi(\mathbf{s},\mathbf{a}) d\mathbf{a} - \int \nabla_\theta \log \pi_\theta(\mathbf{a}|\mathbf{s}) \nabla_\mathbf{a} \pi_\theta(\mathbf{a}|\mathbf{s}) \nabla_\mathbf{a} Q^\pi(\mathbf{s},\mathbf{a}) d\mathbf{a} \tag{25}$$

$$= \int \nabla_\theta \nabla_\mathbf{a} \pi_\theta(\mathbf{a}|\mathbf{s}) \nabla_\mathbf{a} Q^\pi(\mathbf{s},\mathbf{a}) d\mathbf{a} - \overbrace{\nabla_\theta \pi_\theta(\mathbf{a}|\mathbf{s}) \nabla_\mathbf{a}^2 Q^\pi(\mathbf{s},\mathbf{a})|_{\mathbb{R}^n}}^{0} - \int \nabla_\theta \log \pi_\theta(\mathbf{a}|\mathbf{s}) \nabla_\mathbf{a} \pi_\theta(\mathbf{a}|\mathbf{s}) \nabla_\mathbf{a} Q^\pi(\mathbf{s},\mathbf{a}) d\mathbf{a} \tag{26}$$

$$= \int \nabla_\theta \nabla_\mathbf{a} \pi_\theta(\mathbf{a}|\mathbf{s}) \nabla_\mathbf{a} Q^\pi(\mathbf{s},\mathbf{a}) d\mathbf{a} - \int \nabla_\theta \log \pi_\theta(\mathbf{a}|\mathbf{s}) \nabla_\mathbf{a} \pi_\theta(\mathbf{a}|\mathbf{s}) \nabla_\mathbf{a} Q^\pi(\mathbf{s},\mathbf{a}) d\mathbf{a} \tag{27}$$

$$= \int \nabla_\theta \nabla_\mathbf{a} \pi_\theta(\mathbf{a}|\mathbf{s}) \nabla_\mathbf{a} Q^\pi(\mathbf{s},\mathbf{a}) d\mathbf{a} - \int \pi_\theta(\mathbf{a}|\mathbf{s})^{-1} \nabla_\theta \pi_\theta(\mathbf{a}|\mathbf{s}) \nabla_\mathbf{a} \pi_\theta(\mathbf{a}|\mathbf{s}) \nabla_\mathbf{a} Q^\pi(\mathbf{s},\mathbf{a}) d\mathbf{a} \tag{28}$$

$$= \mathbb{E}_{\mathbf{a} \sim \pi}[(\pi_\theta(\mathbf{a}|\mathbf{s})^{-1} \nabla_\theta \nabla_\mathbf{a} \pi_\theta(\mathbf{a}|\mathbf{s}) - \pi_\theta(\mathbf{a}|\mathbf{s})^{-2} \nabla_\theta \pi_\theta(\mathbf{a}|\mathbf{s}) \nabla_\mathbf{a} \pi_\theta(\mathbf{a}|\mathbf{s})) \nabla_\mathbf{a} Q^\pi(\mathbf{s},\mathbf{a})] \tag{29}$$

$$= \mathbb{E}_{\mathbf{a} \sim \pi}[\nabla_\theta \nabla_\mathbf{a} \log \pi_\theta(\mathbf{a}|\mathbf{s}) \nabla_\mathbf{a} Q^\pi(\mathbf{s},\mathbf{a})] \tag{30}$$

where the step in Eq. 26 comes from integration by parts, and we assume that $\nabla_\theta \pi_\theta(\mathbf{a}|\mathbf{s})$ goes to 0 as $\mathbf{a}$ goes to infinity. The last step can be derived in reverse by expanding $\nabla_\mathbf{a} \log \pi_\theta$ as $\pi_\theta^{-1} \nabla_\mathbf{a} \pi_\theta$ and applying the product rule.

## A.3. Variance Update for SVG(0) vs. WPO

$$\Delta\theta = \frac{1}{\sigma} \mathbb{E}\left[(a - \mu)\nabla_a Q(a)\nabla_\theta \sigma\right]$$

$$= \frac{\nabla_\theta \sigma}{\sigma} \int_a \pi(s,a)\left[(a-\mu)\nabla_a Q(a)\right]$$

$$= -\frac{\nabla_\theta \sigma}{\sigma} \int_a \nabla_a\left[\pi(s,a)(a-\mu)\right] Q(s,a)$$

$$= -\frac{\nabla_\theta \sigma}{\sigma} \int_a \left[\nabla_a \pi(s,a)(a-\mu) + \pi(s,a)\nabla_a(a-\mu)\right] Q(s,a)$$

$$= -\frac{\nabla_\theta \sigma}{\sigma} \mathbb{E}_\pi[Q(s,a)\left[(a-\mu)\nabla_a \log \pi(s,a) + 1\right]]$$

$$= -\frac{\nabla_\theta \sigma}{\sigma} \mathbb{E}_\pi\left[Q(s,a)\left[-(a-\mu)(a-\mu)\sigma^{-2} + 1)\right]\right]$$

$$= -\nabla_\theta \sigma \mathbb{E}_\pi\left[Q(s,a) \underbrace{(a-\mu)^2(-\sigma^{-3} + \sigma)}_{-\nabla_\sigma \log \pi}\right]$$

$$= \mathbb{E}_\pi[Q(s,a)\nabla_\sigma \log \pi(s,a)\nabla_\theta \sigma].$$

We note that the last equation is the standard expected policy gradient with respect to the gradients flowing through the variance term. (If we are updating both mean and variance, as is typically, we just add the contributions with respect to the mean to get the full policy gradient update.)

---

**Algorithm 1** WPO with Replay and n-step TD critic learning for multi-dimensional Gaussian Policy

---

**Require:** Initialize actor $\pi_\theta(\boldsymbol{a}|\boldsymbol{s})$, critic $Q_w(\boldsymbol{s}, \boldsymbol{a})$, target actor $\pi_{\bar{\theta}}(\boldsymbol{a}|\boldsymbol{s})$, and target critic $Q_{\bar{w}}(\boldsymbol{s}, \boldsymbol{a})$ with parameters $\theta$, $w$, $\bar{\theta} \leftarrow \theta$, and $\bar{w} \leftarrow w$ respectively. Initialize replay buffer $D$.

1: **for** each episode **do**
2:     Initialize state $\boldsymbol{s}_0$
3:     **for** $t = 0$ to $T - 1$ **do**
4:         Select action $\boldsymbol{a}_t \sim \pi_\theta(\boldsymbol{a}|\boldsymbol{s}_t) = \mathcal{N}(\boldsymbol{\mu}_\theta(\boldsymbol{s}_t), \boldsymbol{\Sigma}(\boldsymbol{s}_t))$
5:         Execute action $\boldsymbol{a}_t$ and observe reward $r_t$ and next state $\boldsymbol{s}_{t+1}$
6:         Store transition $(\boldsymbol{s}_t, \boldsymbol{a}_t, r_t, \boldsymbol{s}_{t+1})$ in replay buffer $D$
7:         **if** length($D$) $\geq$ batch_size **then**
8:             Sample a mini-batch of transitions $(\boldsymbol{s}_j, \boldsymbol{a}_j, r_j, \boldsymbol{s}_{j+1})$ from $D$
9:             Sample n-step transitions $(\boldsymbol{s}_{j+k}, \boldsymbol{a}_{j+k}, r_{j+k})$ for $k = 1, \ldots, n$ or until the end of the episode is reached.
10:            Calculate n-step TD target (using target critic):
11:            $G_{j:j+n} = \sum_{k=0}^{n-1} \gamma^k r_{j+k} + \gamma^n \frac{1}{N} \sum_i Q_{\bar{w}}(\boldsymbol{s}_{j+n}, \boldsymbol{a}'_{j+n,i})$ (or greedy)
                where $\boldsymbol{a}'_{j+n,i} \sim \pi_{\bar{\theta}}(\boldsymbol{a}|\boldsymbol{s}_{j+n}) = \mathcal{N}(\boldsymbol{\mu}_{\bar{\theta}}(\boldsymbol{s}_{j+n}), \boldsymbol{\Sigma}_{\bar{\theta}}(\boldsymbol{s}_t))$
12:            **Update critic:**
13:            $w \leftarrow w - \beta_Q \nabla_w \frac{1}{2}(Q_w(\boldsymbol{s}_j, \boldsymbol{a}_j) - G_{j:j+n})^2$
14:            Update actor (with action sampling):
15:            **for** $i = 1$ to $N$ (Number of action samples) **do**
16:                Sample action $\boldsymbol{a}'_j \sim \pi_\theta(\boldsymbol{a}|\boldsymbol{s}_j) = \mathcal{N}(\boldsymbol{\mu}_\theta(\boldsymbol{s}_j), \boldsymbol{\Sigma}(\boldsymbol{s}_j))$
17:                Calculate policy gradient for the sampled action:
18:                $\textcolor{red}{g_i = \bar{\nabla}_\theta \nabla_a \log \pi_\theta(\boldsymbol{a}'_j|\boldsymbol{s}_j) \nabla_a Q_w(\boldsymbol{s}_j, \boldsymbol{a}'_j)}$ // **WPO Update**
                    where $\bar{\nabla}_{\boldsymbol{\mu}} = \boldsymbol{\Sigma} \nabla_{\boldsymbol{\mu}}$ and $\bar{\nabla}_{\boldsymbol{\Sigma}} = \frac{1}{2} \boldsymbol{\Sigma} \nabla_{\boldsymbol{\Sigma}}$
19:            **end for**
20:            **Update actor using the average gradient:**
21:            $\theta \leftarrow \theta + \beta_\pi \left[ \frac{1}{N} \sum_{i=1}^N g_i - \alpha \nabla_\theta D_{\mathrm{KL}}[\pi_{\bar{\theta}} || \pi_\theta] \right]$
22:            Update target networks:
23:            **if** $t$ mod TARGET_PERIOD $= 0$ **then**
24:                Update target networks (hard update):
25:                $\bar{\theta} \leftarrow \theta, \bar{w} \leftarrow w;$
26:            **end if**
27:        **end if**
28:        $\boldsymbol{s}_t \leftarrow \boldsymbol{s}_{t+1}$
29:    **end for**
30: **end for**

---

# B. Experiment Hyperparameters

All algorithms, WPO, MPO, DDPG and SAC are implemented in the same codebase.

## B.1. DeepMind Control Suite

We use a single set of hyperparameters for each learning algorithm across all control suite environments.

We use a longstanding implementation of MPO which accordingly has well optimised hyperparameters. For WPO, DDPG and SAC, we sought hyperparmeters which worked well across all control suite tasks. Drawing some values from the wider literature on continuous control, we performed a hyperparemeter sweep. We arrived at parameters which performed well across the suite of tasks.

Our focus is on the comparison of algorithms rather than the search for the best hyperparameters.

We note that improved performance can be attained though environment-specific hyperparameter tuning. We found this to be particularly true for Soft-Actor Critic. Maintaining hyperparameter settings across environments helps us investigate variation due to learning algorithm differences.

*Table 1.* Common Hyperparameters

| Hyperparameter | Value |
|---|---|
| Actor Network Hidden Layer Sizes | (256, 256, 128) |
| Critic Network Hidden Layer Sizes | (512, 512, 256) |
| Optimizer | ADAM |
| Actor Learning Rate | $3 \times 10^{-4}$ |
| Critic Learning Rate | $3 \times 10^{-4}$ |
| Activation Function | ELU (Clevert et al., 2016) |
| Discount Factor ($\gamma$) | 0.99 |
| Target Network Update Period | Every 100 updates |
| Samples-per-insert (SPI) | 32 |
| Batch Size | 256 |
| n (for TD(n) loss calculation) | 5 (Except SAC, which used $n = 3$) |
| Replay Buffer Size | $2 \times 10^6$ |
| Trajectory Length for Network Update | 10 |

*Table 2.* WPO Hyperparameters

| Hyperparameter | Value |
|---|---|
| $\alpha_\mu$ | log(2) |
| $\alpha_\Sigma$ | 10000 |
| Actions Sampled per Update | 30 |

*Table 3.* DDPG Hyperparameters

| Hyperparameter | Value |
|---|---|
| (Fixed) Policy Variance ($\sigma$) | 0.3 |

*Table 4.* MPO Hyperparameters

| Hyperparameter | Value |
|---|---|
| $\alpha_\mu$ (initial) | 100 |
| $\alpha_\Sigma$ (initial) | 1000 |
| Initial Log Temperature ($\log \eta$) | 10 |
| $\epsilon$ | 0.1 |
| $\epsilon_\mu$ | $5 \times 10^{-3}$ |
| $\epsilon_\Sigma$ | $10^{-6}$ |
| Actions Sampled per Update | 30 |

*Table 5.* SAC Hyperparameters

| Hyperparameter | Value |
|---|---|
| Initial $\alpha$ | 0.0001 |
| Minimum $\alpha$ | $1 \times 10^{-8}$ |
| Final Policy Network Activation Function | $\tanh$ |
| Polyak Average Coefficient ($\tau$) | 0.005 |
| Target Entropy | $-|\mathcal{A}|$ |
| Maximum Policy Variance | $\exp(4)$ |
| Minimum Policy Variance | $\exp(-10)$ |

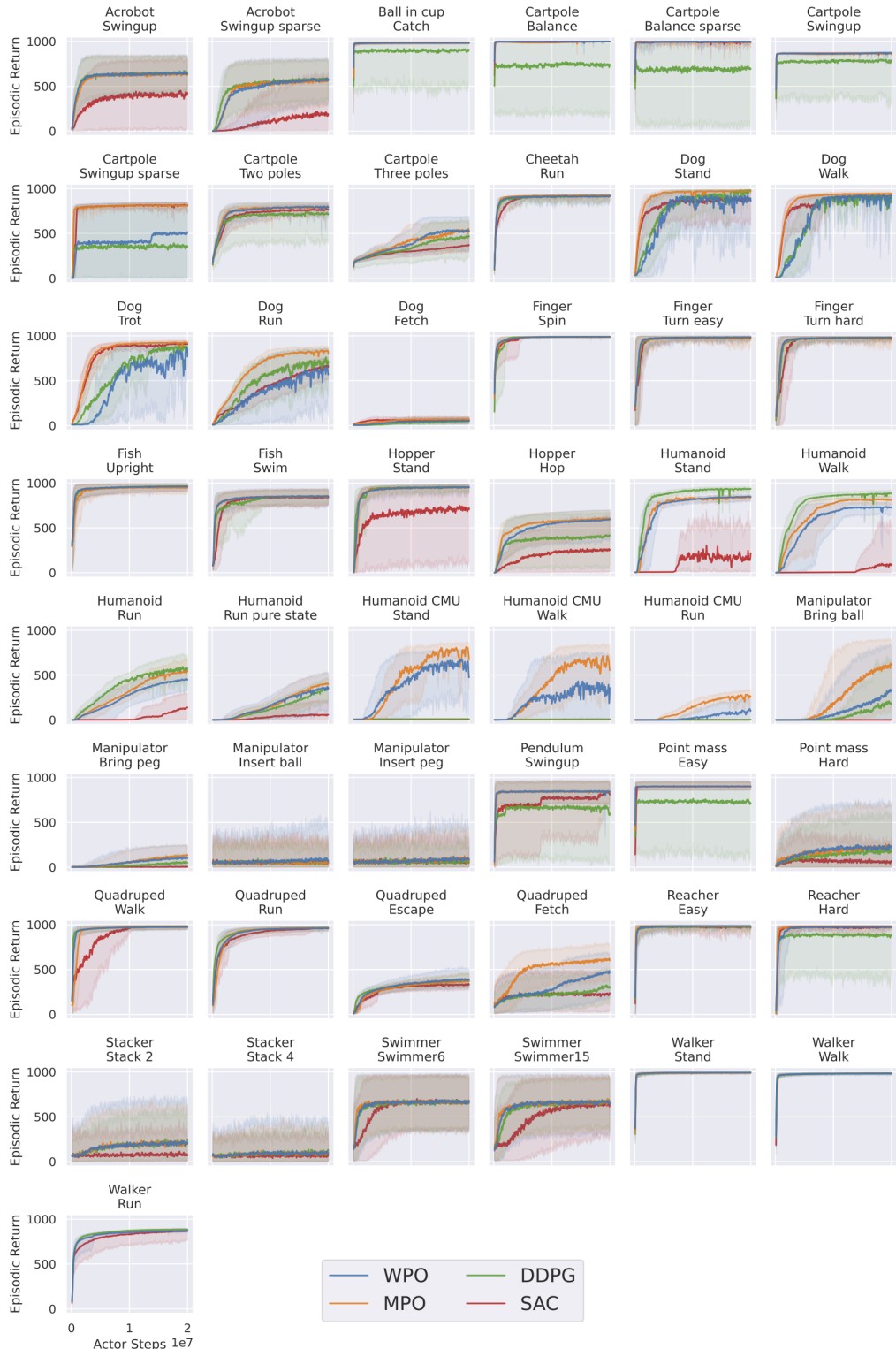

*Figure 7.* Results on all DeepMind Control Suite tasks for WPO, MPO, DDPG and SAC. The bold lines denote average returns over 10 seeds on evaluation episodes. The shaded region spans minimum and maximum returns.

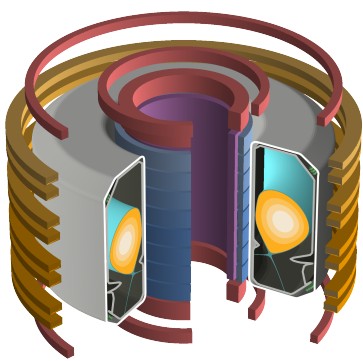

*Figure 8.* Schematic of the TCV tokamak, including cutout showing 2-D profile of plasma contour

## B.2. Combined Tasks

In order to combine the rewards across component environments in the combined tasks we apply a SmoothMin operation. This is the application of a SmoothMax operation with a negative value of $\alpha$.

$$\text{SmoothMax}(x_{1...n}, \alpha) = \frac{\sum_{i=1}^{n} x_i e^{\alpha x_i}}{\sum_{i=1}^{n} e^{\alpha x_i}}. \tag{31}$$

The SmoothMin operation is non-linear. To make the $\alpha$ hyperparameter easier to tune and maintain across differing choices of component environment, we scale the rewards such that they are in the unit interval by dividing by the highest reward observed from any agent in a singleton environment.

## B.3. Fusion Task

Fig 8 depicts the TCV tokamak, including the control coils, vacuum-vessel cross section and a notional plasma inside the domain. The TCV tokamak has 19 control coils, composed of 16 poloidal field coils, 2 ohmic coils, and 1 "fast" coil. The measurements observed by the agent for these experiments consist of 38 magnetic field probes, 34 magnetic flux loops, and 19 coil current measurements, one for each loop. We simulate the evolution of the tokamak using the FGE Grad-Schafranov simulator (Carpanese, 2021). For more details of the training setup, please see (Degrave et al., 2022) and (Tracey et al., 2024).

The task shown in Section 6.2 is based on the shape_70166 task from (Tracey et al., 2024). The primary goal of the task is for the plasma last closed flux surface to align with the control points, and also for the current in the plasma to be maintained at a reference value (not depicted). There are some additional reward terms to regularize the policy and help hardware transfer, specifically penalizing out-of-bounds voltage commands, keeping the ohmic coil currents close together, and keeping the poloidal coil currents away from zero. The results shown in Fig. 9 use a version of the task with tighter demands on the shape and plasma current accuracy. These changes are similar to those discussed by Tracey et al. (2024) with regards to reward shaping. We also show in Fig. 10 results from the original task in (Tracey et al., 2024). The specifics of the reward components and combiner is shown in Table 6, please see Sec. 3 of (Tracey et al., 2024) and in particular Eq.2 and Eq. 3 for how these terms combine into a scalar reward. In both figures, MPO and WPO are run with the same set of hyperparameters, must notably with a hard constraint on the KL regularization with $\epsilon_\mu = 5e - 5$ and $\epsilon_\sigma = 1e - 7$. Note that the KL constraint on the variance, $\epsilon_\sigma$, is set to the same value for WPO and MPO. So while the value is tighter than typically seen, it is not responsible for the difference in policy convergence rates between the two algorithms.

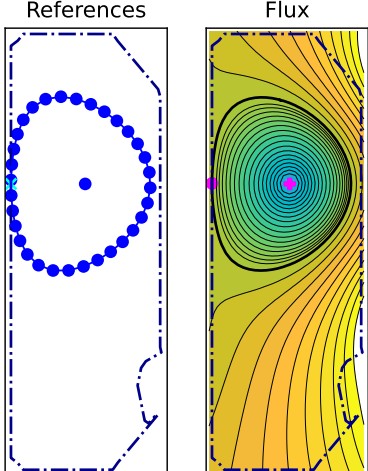

*Figure 9.* The reference control points and example flux field for the `shape_70166` task. The last closed flux surface target points are shown in blue on the left, with the limit point location in cyan. An example magnetic flux field matching these control points in shown on the left, with the limit location in magenta and the LCFS in black

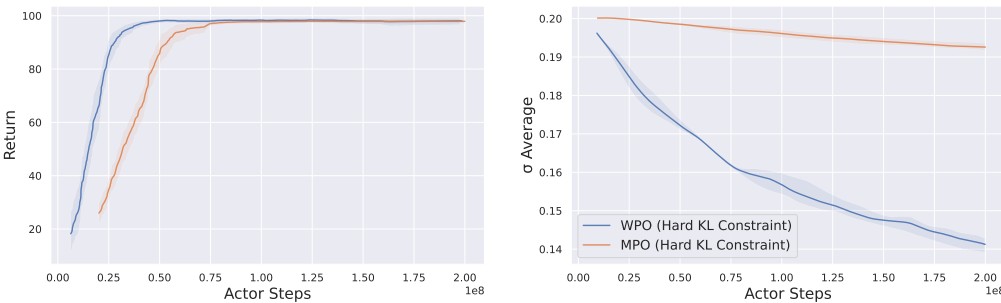

*Figure 10.* Return and policy average standard-deviation evolution throughout training on the `shape_70166` task from (Tracey et al., 2024)

## C. Additional Results and Ablations

### C.1. Complete DeepMind Control Suite Results

In Fig. 7, we show results on all 49 Control Suite tasks (excluding the LQR domain, which suffered from numerical issues in simulation) for WPO and baseline algorithms, providing a more complete picture of the results in Fig 4. It can be seen that WPO is competitive with MPO on nearly all tasks, while there are numerous tasks for which DDPG and SAC struggle, especially those on the Cartpole and Point mass domains for DDPG and Acrobot and Humanoid tasks for SAC. Notably, even on those tasks where DDPG outperforms other methods, WPO still converges eventually, while many tasks for with WPO performs well, DDPG does not reliably converge. We note that the performance of WPO on some domains is sensitive to modeling choices. We explore these choices below.

### C.2. MPO Hyperparameter Comparison

To ensure that we are making a fair comparison between WPO and the strongest possible baseline on DeepMind Control Suite, we investigated several different hyperparameter settings for MPO, with results shown in Fig. C.2. In addition to the hyperparameters in Table 4, we also ran MPO with the same hyperparameters as WPO – replacing the hard KL constraint with a soft constraint and relaxing the KL regularization on the mean. We additionally tried adding the variance rescaling trick described in Sec. 5. As can be seen in the figure, these modifications did not significantly affect the performance of

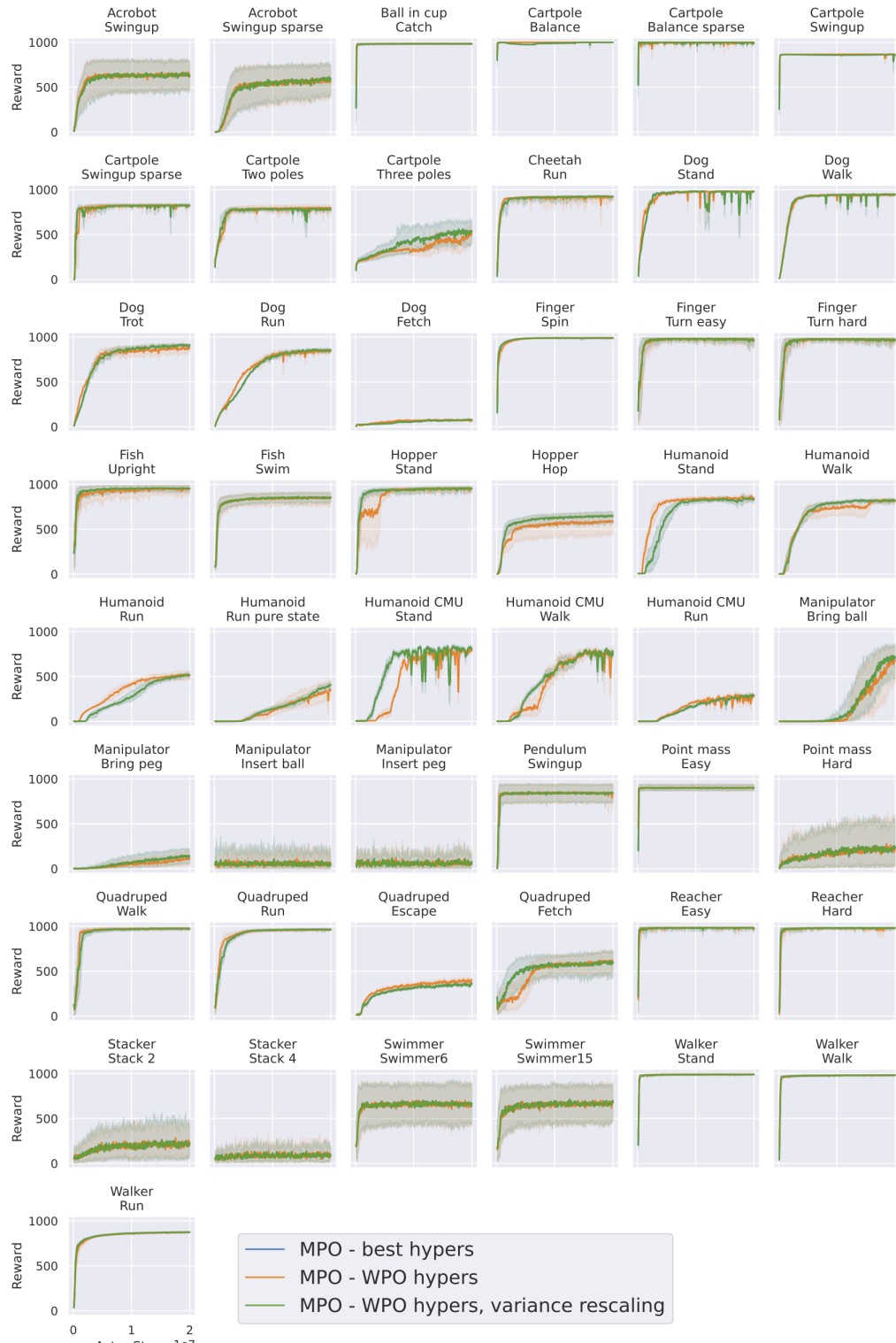

*Figure 11.* Results on all DeepMind Control Suite tasks for MPO with different hyperparameters: optimal settings tuned for Control Suite, settings matching WPO, and settings matching WPO with variance rescaling. The performance of MPO is not significantly impacted by these choices.

| Component | Precise | | Standard | |
|---|---|---|---|---|
| | Good | Bad | Good | Bad |
| LCFS Distance (m) | 0.001 | 0.01 | 0.005 | 0.05 |
| Plasma Current (A) | 100 | 2000 | 500 | 30000 |
| Poloidal away from 0 (A) | 100 | 50 | 100 | 50 |
| Normalized Voltages in Bounds | 0 | 1 | 0 | 1 |
| Ohmic Coils Close (A) | 50 | 1050 | 50 | 1050 |
| Combiner | -3 | | -0.5 | |

*Table 6.* Fusion task reward components and scaling parameters

MPO. With the WPO hyperparameters, the Humanoid CMU and Hopper tasks were slightly slower to learn for some tasks, but adding the variance rescaling largely restored the performance. This gives us high confidence that our baseline is as strong as possible, and that any difference in performance cannot be attributed to arbitrary hyperparameter differences.

**C.3. Squashing Functions and Network Activations for WPO**

There are various possible sources of numerical instability unique to WPO which we investigate. One is that, for certain tasks where the action-value function changes rapidly, the gradient $\nabla_\mathbf{a} Q^\pi(\mathbf{s}, \mathbf{a})$ might blow up, making the gradients unstable. Fortunately, there is a fix for this which is still a principled approach to approximating Wasserstein gradient flows. In Neklyudov et al. (2023), it was shown that much of the theory in Sec. 2.2 and 2.3 can be extended to the c-Wassserstein distance:

$$W_c^2(\pi_0, \pi_1) = \inf_{\rho \in \Gamma(\pi_0, \pi_1)} \int \rho(\mathbf{a}, \mathbf{b}) c(\mathbf{a} - \mathbf{b}) d\mathbf{a} d\mathbf{b} \tag{32}$$

where $c$ is some convex function. In this case, the appropriate PDE for the c-Wasserstein gradient flow to minimize $\mathcal{J}$ becomes

$$\frac{\partial \pi}{\partial t} = -\nabla_\mathbf{a} \cdot \left( \pi \nabla c^* \left( -\nabla_\mathbf{a} \frac{\delta \mathcal{J}}{\delta \pi} \right) \right) \tag{33}$$

where $c^*$ is the convex conjugate of $c$. This means that applying a nonlinear squashing function to the velocity field can still result in dynamics that minimize the functional of interest.

If we follow the same derivation of the parametric update in Sec. 2.3 and A.2 for the c-Wasserstein distance, the derivation is largely unchanged, but the term $\nabla_\mathbf{a} Q^\pi(\mathbf{s}, \mathbf{a})$ is replaced with $\nabla c^*(\nabla_\mathbf{a} Q^\pi(\mathbf{s}, \mathbf{a}))$. This means applying a nonlinear function to the gradient of the action-value function still results in a principled update, which may be useful for numerical stability in cases where the action-value function changes rapidly. To investigate the impact that this has on the performance of WPO on the DeepMind Control Suite, we choose $\nabla c^*(\mathbf{a}) = \mathbf{a}^{1/3}$, where the cube root is applied elementwise. This is a natural choice of squashing function as it is smooth, odd, and doesn't saturate.

Another possible source of numerical instability comes from the choice of neural network activation. The exponential linear unit (ELU) (Clevert et al., 2016) has a discontinuity in the second derivative, which may cause issues with WPO as the second derivative appears in the update. We try changing both the actor and critic networks to use sigmoid linear units (SiLU) (Elfwing et al., 2018). We present the results of both changing the network nonlinearity and adding a cube root squashing function to the action value gradient in Fig. 12

On most tasks, the choice of nonlinearity and squashing function does not make a significant difference. On a number of tasks such as Cartpole - Swingup sparse and tasks on the Dog domain, both changing the nonlinearity and adding a squashing function seem to improve performance, but on the Humanoid CMU domain they potentially harm performance. Therefore, it's difficult to conclude that any of these choices decisively improve performance, but it is important to be aware that some of the results presented in the main paper can be sensitive to these choices.

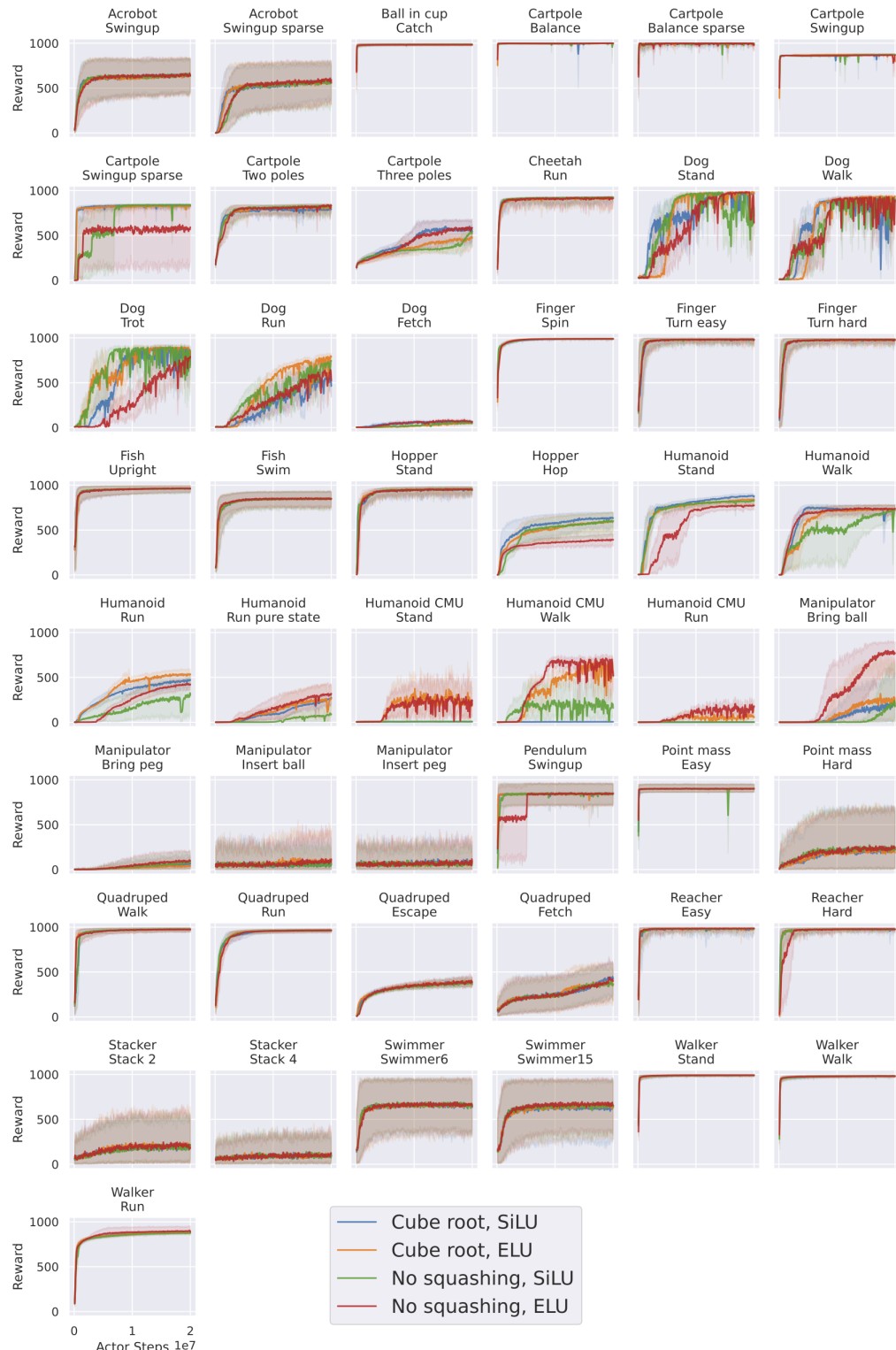

*Figure 12.* Results on all DeepMind Control Suite tasks for WPO with different activation functions (ELU vs. SiLU) and with and without applying a cube root elementwise to the action-value gradients. The bold lines denote average returns over 3 seeds on evaluation episodes. The shaded region spans minimum and maximum returns.

