# OpenReview forum: "Wasserstein Policy Optimization"
_ICML.cc/2025/Conference — ICML 2025 poster_

### Official Review · Reviewer_PGXN · 2025-03-06

**Overall Recommendation:** 1

**Summary:**

The paper applies Wasserstein gradient flows to reduce the parameter space, and this way obtain a closed-form update rule. It lifts the necessity of using the reparameterization trick in stochastic policy learning. The method merges the strengths of the policy gradient approaches that work on sampled evaluations of the action-value and deterministic policy gradients that work on a single steepest-ascent direction of the action-value. It can generate a vector field towards an increasing direction.

**Claims And Evidence:**

I am having hard time to conclude from Figures 3 and 4 that the proposed method improves the state of the art. The alternative approaches consistently outperform the proposed WPO approach. This is a rather strong evidence against the central claim of the paper: Training with a Wasserstein gradient flow facilitates learning.

**Essential References Not Discussed:**

The paper does cite the most important papers with respect to the chosen problem formulation. With this regard, I do not have concerns. However, I will reiterate that problem formulation has issues.

**Experimental Designs Or Analyses:**

The chosen baselines make sense. However, the DMC results do not match the ones I know from the literature and from my own experience. For instance, in Humanoid Walk, I am able to reach much higher performances with vanilla SAC in much less actor steps than what is reported. I suspect that this is because the authors train their networks with a single critic. There are also some additional hyperparameter choices which are uncommon, such as using 5-step temporal difference and using hard target network updates in intervals instead of Polyak averaging. All in all, I am not able to conclude that the reported results are informative about the match of the claims to evidence.

**Methods And Evaluation Criteria:**

The adopted methodology to build a deep actor-critic algorithm, the built experiment, pipeline, and the used performance scores make sense. There are issues about the choice of the baselines as well as the implementation details of the proposed algorithm. I give more details under Experimental Designs or Analyses.

**Other Comments Or Suggestions:**

Post-rebuttal update:

Having completed the rebuttal and discussions, I am still missing the concrete problem for which a solution is sought. The paper describes the problem as:

"Adding in stochasticity can be difficult to tune, and extensions that learn the variance (Heess et al., 2015; Haarnoja et al., 2018) rely on the reparameterization trick, which limits the class of policy distributions that can be used."

My grade stems for a serious doubt about the significance of this problem and the claim that the solution addresses it. The proposed solution does work, however the experiment setup does not represent the state of the art in the sense that the decade-old techniques to mitigate the harmful effects of the Bellman target are not used. The outcome is then a set of half-working models, one new, the other two are baselines. The new one is working tiny bit better than others. The questions would such a performance difference be visible when all the commonplace training stabilization techniques were used.

The authors clarify in their response is that the results are shown in the proprioceptive setup. I have first-hand experience to achieve much better results in much less environment interactions using multiple methods all of which are well-known in the literature. Under these circumstances I cannot conclude that the paper makes a concrete scientific contribution. Hence I maintain my score.

**Other Strengths And Weaknesses:**

While the idea of using Wasserstein Gradient Flows to create a sweet spot between PPO-like and DDPG-like algorithms is an interesting idea, the proposed solution is straightforward in the sense that it does not yield downstream scientific challenges that would contribute to the advancement of the field. The implementation of the idea also has severe flaws I pointed out above.

**Questions For Authors:**

Why is $\nabla_\theta \pi(s_t)$ a Jacobian but not the gradient?

Are the DMC results obtained on the visual (pixel-based) or proprioceptive control setting? I interpret them as the latter in my evaluation above. I can update it based on the answer here.

**Relation To Broader Scientific Literature:**

The paper takes the Maximum A-Posterior Policy Optimization (MPO)  and DDPG as the prior work on which it brings an improvement. While I agree that MPO is an appropriate representative of stochastic policy gradient approaches, I do not think DDPG represents the state of the art. Its TD3 variant works significantly better. I would expect discussion about how the significant problem and the proposed method comes together with the established state of the art that uses DPG on an ensemble of critics, which essentially addresses quite many of the issues targeted by the WPO solution. For example the REDQ method is known to bring significant reduction to the gradient estimator variance while also correcting the overestimation bias. I would definitely expect a comparison against it to draw a conclusion about the value of the solution. The authors may see the following detailed benchmarking study to have an impression about what is possible with the existing approaches:

Nauman et al., Overestimation, Overfitting, and Plasticity in Actor-Critic: the Bitter Lesson of Reinforcement Learning, ICML, 2024

**Theoretical Claims:**

I checked especially the functional derivative proof in Appendix A.1 in detail and it looks in order.

---

> ### Author Rebuttal · Authors · 2025-03-31
>
> We thank the reviewer for their comments and hope to convince them of the merits of our approach.
> 1. With regards to the performance of WPO relative to other baseline methods: while Figs. 3 and 6 show that WPO is competitive over the DM control suite, our main performance claims are highlighted in Figs. 4 and 5 that show that, as conjectured, WPO especially shines in domains with relatively high-dimensional action spaces. We would also dispute the characterization that “the alternative approaches consistently outperform the proposed WPO approach”. As we state in the paper, no single approach consistently outperforms the others across all tasks. Even on DeepMind Control Suite, there are tasks for which WPO performs the best, such as Cartpole - Three Poles, and many others where it matches state-of-the art. We also think the reviewer may misunderstand the main claim of the paper. We are *not* claiming that “training with a Wasserstein gradient flow facilitates learning” in all cases. We are claiming that WPO is novel, fills in a much needed gap in the space of actor-critic algorithms (how to generalize DPG-like updates to arbitrary stochastic policies without reparameterization) and, with only minimal effort, can be adapted into a performant deep RL algorithm.
> 2. In response to the issues raised in “Experimental Designs and Analyses”: we wanted to show that the WPO gradient (Eq. 6) can be dropped into existing deep RL methods with minimal modification, and to do a fair comparison we wanted the baselines to be trained in as similar a fashion as possible. Our results use the same hyperparameters per algorithm across all tasks. Many deep RL algorithms are highly sensitive to small details in how they are trained. Thus we may exactly match the absolute best performance of each baseline method. However, where possible, we made sure to compare against numbers in the literature to make sure our comparison is fair - for instance, we both ran the open-source Dopamine implementation of SAC ourselves and added baseline numbers from the literature.
> Since submission, we implemented SAC in the same framework that was used for the other experiments, and will include those results in the final camera-ready paper. We are also running DDPG with 1-step updates instead of 5-step updates for comparison and can include those results in the appendix, although they do not look appreciably better than what we have already included. In general, while it may be possible to tune the hyperparameters to get better baseline performance, it may also be possible to tune WPO hyperparameters to get better performance, so we believe the comparison provided in the paper is quite fair.
> 3. Re:TD3 and other possible baseline methods, we are primarily interested in the WPO update itself (Eq. 6), and compare like-for-like with algorithms like MPO. The algorithm we arrive at in this paper which we call “WPO” is one of an enormous number of possible algorithms that use the update in Eq 6. TD3 is an extension of DDPG that has some additional algorithmic ideas like target policy smoothing that can also be combined with other algorithms. It would be lovely to see future work decomposing these separable ideas and combining them in novel ways, to understand the individual contributions of all these ideas separately, rather than just comparing complete agents with somewhat-arbitrary combinations. Maybe WPO with TD3-like tricks will outperform TD3, maybe not, but that is out of scope of this particular paper.
> 4. When you say “the proposed solution is straightforward in the sense that it does not yield downstream scientific challenges that would contribute to the advancement of the field” - we do not understand if this is a criticism or a compliment. A solution which is straightforward and does not create future challenges is a good thing, is it not?
> 5. “Why is ∇_θπ(s_t) a Jacobian but not the gradient?” - ∇_θπ(s_t) is only a Jacobian in the case of DPG, because in this case π(s_t) is a deterministic vector-valued mapping from state vector to action vector. In the case of WPO or standard policy gradient, π(a_t|s_t) is a scalar-valued probability density and thus ∇_alogπ(a_t|s_t) is a vector of the same shape as π(s_t) in DPG.
> 6. The DMC results are based on proprioceptive control, not pixel-based control.
>
> We hope that this resolves any lingering issues with the paper. We want to emphasize again that our goal was not to show that we have the right combination of tricks and hyperparameter settings to achieve optimal performance across all possible tasks. We wanted to show that an open question in continuous control - how to train arbitrary stochastic policies with updates that exploit the action-value gradient - has a clear and elegant answer using approximations to Wasserstein gradient flows, and make a first foray into translating that into a practical deep RL algorithm. We hope you will agree that that is a significant contribution worth publishing at ICML.

---

> > ### Comment · Reviewer_PGXN · 2025-04-05
> >
> > Thanks for the reply. Overall I am not able say I am convinced.
> >
> > 1. and 2.) The replies verify my interpretation that the proposed method is not demonstrated to improve performance and it is not sufficiently challenged. I am completely fine by this if any other theoretical or empirical benefit is demonstrated (e.g. a tighter convergence proof, a generalization performance guarantee, a new algorithmic design that follow unconventional steps of deductive reasoning, a steeper learning curve, or a reduced computation time). The issue is not about whether hyperparameters are tuned. It is about a mismatch of the given answer and the posed scientific question.
> >
> > 3.) If DDPG qualifies as a baseline, TD3 (its truly working variant) definitely does. The rebuttal answer to this point justifies the issues I raised in my original review about the adopted scientific method. WPO update itself is a worthwhile scientific problem if a pain point of the state of the art can be demonstrated. I do not see any effort for doing this and also any improvement in the results for addressing any.
> >
> > 4.) For example, showing a performance improvement by increasing model capacity or access to more information is straightforward. There are many ways to put together different ideas. Some are avoided as being straightforward, meaning that the consequences are predictable. Hence, it does not raise new scientific questions or attract interest in different aspects of the same problem. This is what I meant and I am missing a core scientific problem and a solution that shows a clear practical or conceptual benefit. I read the paper in good enough detail, so I don't think another summarization will help there.
> >
> > 5. and 6.) Thanks, understood.

---

### Official Review · Reviewer_L61D · 2025-03-10

**Overall Recommendation:** 4

**Summary:**

This paper naturally and clearly derives a natural gradient version of the stochastic policy extension of deterministic policy gradient from the perspective of Wasserstein gradient flow, proposes a practical implementation method, and conducts detailed tests on DeepMind Control Suite and scientific control tasks, achieving comparable results.

**Claims And Evidence:**

yes.

**Essential References Not Discussed:**

Enough

**Experimental Designs Or Analyses:**

The authors conduct extensive experiments on the widely used DMC and show that the proposed method is comparable to the commonly used methods.

**Methods And Evaluation Criteria:**

yes.

**Other Comments Or Suggestions:**

In line 685, there is a missing right bracket before $\nabla_a Q^\pi(s,a)$.

I really appreciate that authors conducted comparative experiments on the magnetic control problem, which is extremely valuable. Since the simulator is closed-source and few groups used it, it is difficult for other RL researchers to reproduce the experimental results. If authors have plans to make the experimental code open source, it will yield more benefits.

**Other Strengths And Weaknesses:**

The overall writing of the paper is clear, the overall motivation is well supported, and the relationship with previous work is fully discussed. My main concern about the paper is that the experimental performance results are relatively comparable with baseline methods mainly focusing on the comparison with the MPO algorithm, and the selected comparison methods lack popular algorithms such as PPO and TD3. The paper omits the $d^\pi(s)$ term when deriving the formula, and I think it is better to add it. In line 358 of the left column, the author says that off-policy state data can be used, but this will cause $d^\pi(s)$ to be outdated, making the gradient direction not the direction in which $\mathcal{J}$ decreases the fastest.

**Questions For Authors:**

Many current works like Zhang et al. (2018) choose the reparameterization trick because the method is convenient for sampling actions. The authors emphasize in the paper that the main difference between WPO and other methods is whether to use parameterization. Can authors give some more practical examples or scenarios to illustrate the necessity of using some difficult-to-reparameterize policies?

In line 229 of the right column, the authors mentioned that for the sampled case, the variance of WPO is 0 when Q is locally linear in the actions. I am a little confused about this because for the method using reparameterization, this property is still enjoyed in this case due to $\frac{\partial Q}{\partial \theta}=\frac{\partial Q}{\partial a}\frac{\partial a}{\partial \theta}=w(s)\frac{\partial a}{\partial \theta}$.

Have the authors considered the application of this method to the diffusion model as a policy representation? The diffusion model can be considered to directly model $\nabla_a \log \pi$. I think this may be interesting.

**Relation To Broader Scientific Literature:**

This paper proposes a new reinforcement learning policy update method, and the paper explains the motivation well from the perspective of Wasserstein gradient.

**Theoretical Claims:**

I have checked most of the proofs but cannot ensure that they are entirely accurate.

---

> ### Author Rebuttal · Authors · 2025-03-31
>
> We thank the reviewer for the comments and helpful suggestions.
> 1. With regards to the state-occupancy term $d^\pi(s)$ in the definition of the gradient of the value function, we agree that including a correction in the off-policy case could potentially improve performance, but want to emphasize that virtually no methods in off-policy RL actually use a correction of this form in practice. Therefore, our algorithm conforms to standard practice and we are confident that the experimental comparisons in the paper are fair. One reason that this term is almost always excluded is that estimating it in continuous state spaces is extremely challenging. It’s also worth highlighting that it has been argued in previous work that excluding this term actually can improve the efficiency of the updates, and it has been shown to be theoretically a sound thing to do (see, e.g., https://arxiv.org/pdf/2302.11381). We will discuss this explicitly in the paper.
> 2. We have fixed the typo in the equation on line 685
> 3. Unfortunately, we do not control the rights to the fusion simulator used in our experiments, and so have no control over open-sourcing it. However, the organization which does own the rights intends to release an open-source version of the simulation code in the long term. We will do our best to encourage them to release this as soon as they can.
> 4. The use of the reparameterization trick becomes much more complicated with distributions with discrete latent variables, like mixtures of Gaussians. The exact SVG(0) update, and the efficiency of this, will be sensitive to the choice of reparameterization. It is difficult to find a practical problem for which mixture-of-Gaussian action distributions are necessary, since Gaussian action distributions work well in practice for most environments investigated here. However, we will extend Section 4 to include a toy example of a mixture of Gaussians policy on a complex loss landscape showing the qualitative difference in dynamics between WPO and standard policy gradient, and Gaussian vs. mixture-of-Gaussian action distributions.
> 5. The discussion of the variance of the linear action-value function case in Section 4 can be improved. First, we will clarify that the WPO gradient is only constant when the variance of the policy is fixed (e.g. only the mean is updated). Second, we will clarify that it is only standard policy gradient for which the variance is finite, while SVG(0) with the appropriate parameterization is identical to WPO, as you noted. Third, we will highlight that the agreement between WPO and SVG(0) is only true for specific choices of parameterization for SVG(0), and provide a concrete example where this identity is broken. For instance, replacing a Gaussian action distribution with an exponential distribution π(a) = exp(a/β)/β (with the natural reparameterization a = β * η) leads to different updates for WPO and SVG(0), even in expectation: the expected natural WPO update with respect to the scale parameters β will then simply be E[ ∇_a Q(a) ], while the SVG(0) update will be E[ ∇_a Q(a) diag(a/β) ], with η a standard exponential with p(η = x) = exp(-x). (If we consider the gradient with respect to the network parameters, both these updates just get multiplied with the Jacobian ∇_θ β, as per standard backprop.)
> 6. We agree that using WPO in combination with diffusion models is a promising future avenue to investigate. Thank you for the suggestion.

---

### Official Review · Reviewer_mryg · 2025-03-11

**Overall Recommendation:** 3

**Summary:**

This paper introduces Wasserstein Policy Optimization (WPO) for continuous-action reinforcement learning. By viewing policy optimization as a Wasserstein gradient flow in the space of distributions, the authors derive a closed-form update that: i) Uses gradients of action values w.r.t. actions (like deterministic policy gradients). ii) Works for any stochastic policy (no reparameterization needed, unlike SVG or SAC). iii) Can be turned into a simple actor-critic algorithm with minimal additional structure. The paper provides theoretical derivations, then extends WPO with practical features (e.g., KL regularization, a Fisher information rescaling) for deep RL. Experiments on the DeepMind Control Suite (including high-dimensional tasks like Humanoid) and a real-world-inspired fusion-control task show that WPO is robust, sometimes outperforming or matching strong baselines (MPO, SAC, DDPG). The authors highlight WPO’s favorable scaling in large action spaces and its ability to converge stably to high-performing solutions.

## update after rebuttal

I thank the authors for their detailed response and additional experimental results. I continue to find the theoretical formulation of Wasserstein Policy Optimization novel and compelling. I also appreciate the effort made to re-run experiments with more seeds and to clarify the challenges involved in evaluating on larger benchmarks.

That said, some of my original concerns remain. The updated results indicate noticeable variance across seeds in several tasks, and some performance drops (e.g., Humanoid CMU Walk, Manipulator: Bring Ball) raise questions about the method’s robustness. Additionally, while I understand the limitations during the rebuttal phase, the lack of evaluation on more realistic high-dimensional environments makes it difficult to fully assess the claimed scalability of WPO.

Overall, I believe the paper presents an interesting direction, and I hope to see future versions strengthen the empirical foundation. I am maintaining my score.

**Claims And Evidence:**

The paper’s main claims—i.e., that WPO (1) combines properties of deterministic and standard policy gradients, (2) applies to arbitrary continuous action distributions, and (3) matches or outperforms strong baselines in large action spaces—are backed by both theoretical derivations and empirical results on benchmark (DeepMind Control Suite) and real-world-like (fusion) tasks. The paper does not appear to make unsupported or overstated claims

**Essential References Not Discussed:**

No

**Experimental Designs Or Analyses:**

**Experiments in DeepMind Control Suite.** From Figure 3, we can see that there is a high variance environments, e.g. Cartpoles, Don Run, Manipulator Bring ball and Humanoid Stand. It should run more seeds to reduce the variance.

**Experiments in Combinded Tasks.** By replicating a single environment (e.g., Humanoid - Stand) multiple times to expand the action space, the authors leverage SmoothMin to aggregate rewards. This approach effectively tests scalability to higher-dimensional controls, but it still falls short of simulating the diversity and complexity of real-world scenarios. It would be good if test WPO in Bi-DexHands tasks with high-dimensional control.

**Methods And Evaluation Criteria:**

The paper evaluates WPO in three ways. (1) the DeepMind Control Suite, a widely used benchmark for continuous control, enabling clear comparisons with established methods. (2) a Magnetic Confinement Fusion domain, which tests real-world relevance and benchmarks WPO against strong baselines like MPO. (3) Combined Tasks, The paper combines multiple copies of a Control Suite environment to form high-dimensional action spaces, revealing WPO’s potential for scaling to large numbers of actions. These three perspectives offer a well-rounded assessment of WPO’s capabilities.

**Other Comments Or Suggestions:**

Do more experiments to show WPO’s validness.

**Other Strengths And Weaknesses:**

**Strengths:**

- Principled Theoretical Foundation
- Combines Advantages of Deterministic and Stochastic Policy Gradients: This allows WPO to leverage value information for efficient updates without the limitations of deterministic policies or the reparameterization trick.

**Weaknesses:**

- High variance in training: From Figure 4, WPO has higher variance compared with MPO and DDPG.
- Not Universally Superior: While WPO performs competitively on many tasks, it does not consistently outperform state-of-the-art algorithms like SAC and MPO across all environments in the DeepMind Control Suite

**Questions For Authors:**

WPO combines the advantages of deterministic and stochastic policy gradients, but why it is not universally superior than the baselines?

**Relation To Broader Scientific Literature:**

The authors propose Wasserstein Policy Optimization (WPO), presenting it as an extension of classic gradient-based methods (e.g., REINFORCE, actor-critic) by exploiting action-value gradients similarly to DPG, yet without restricting the policy class to deterministic forms. Grounded in optimal transport, WPO’s update reflects a continuous steepest descent in policy space, differing from approaches like MPO that exponentiate Q-values rather than relying on a direct gradient. This formulation aligns stochastic and deterministic policy gradients, offering a unified viewpoint and potentially stronger scalability in high-dimensional action spaces.

**Theoretical Claims:**

There are no apparent flaws in the theoretical arguments as presented.

---

> ### Author Rebuttal · Authors · 2025-03-31
>
> We would like to thank the reviewer for their thoughtful comments and favorable review. The reviewers made a few suggestions for improving the paper which we will address:
> 1. We have re-run experiments on the high-variance environments with more seeds to smooth out the learning curves and will add the new results to the paper. The new results look qualitatively similar to those in the original submission, with some amount of noise.
> 2. We appreciate the suggestion for more challenging high-dimensional tasks to investigate such as Bi-DexHands. The state-of-the-art RL methods for bidextrous manipulation often involve many additional steps beyond blank-slate actor-critic methods, such as imitation learning from human data (e.g. ALOHA Unleashed, https://arxiv.org/pdf/2410.13126). This adds an additional layer of complexity for comparison against baselines that goes beyond the scope of what we have the space and time to investigate in this paper, however we will be sure to investigate these sorts of tasks going forward.
>
> With regards to the lack of uniformly superior performance that the reviewer points out, we refer to the rebuttal to reviewer A8Ro for a longer discussion. We hope that this addresses all of the issues the reviewer has and thank them again.

---

> > ### Comment · Reviewer_mryg · 2025-04-02
> >
> > The rebuttal does not sufficiently address key concerns.
> >
> > - The authors claim to have re-run high-variance experiments with more seeds, but no updated figures or metrics are shown. Without evidence, this claim lacks credibility and weakens the empirical foundation.
> >
> > - The justification that dexterous tasks like Bi-DexHands are “beyond the scope” due to imitation learning is inaccurate. Standardized RL benchmarks for Bi-DexHands without imitation do exist (e.g., [DexterousHands](https://github.com/PKU-MARL/DexterousHands)). Given that WPO emphasizes scalability, and the paper already ventures beyond standard tasks (e.g., fusion control), omitting such realistic, high-dimensional environments undermines the generalization claims.
> >
> > I still find the theoretical framework interesting, but the empirical validation is incomplete. I am lowering my score to 3: Weak Accept.

---

> > > ### Author Response · Authors · 2025-04-07
> > >
> > > We apologize for insufficiently addressing the reviewer's concerns, and will do our best to fix this. First of all, we want to emphasize that there may have been a misunderstanding. Given the initial high score for the paper, we understood the reviewer's comments to be helpful suggestions for further improving the paper, not necessary conditions for maintaining the high score. To the two points raised by the reviewer:
> > >
> > > 1. We did in fact do *exactly* what the reviewer requested here - we re-ran high variance experiments with more seeds. We apologize for not sharing more details of the results of those experiments, but want to point out that there is in fact no mechanism for us to share new figures during the rebuttal period - the paper draft cannot be updated and images cannot be included in the rebuttal text. Instead, we can share some subset of the results here as a table, which we hope is enough to convince the reviewer that the results remain qualitatively similar:
> > >
> > > Humanoid CMU: Walk
> > > $$
> > > \begin{array}{ccc}
> > > \\hline
> > > \\mathrm{Total Steps} (1e6) & \\mathrm{Old Reward} & \\mathrm{New Reward} \\\\
> > > \\hline
> > > 2&2.06\pm1.64&2.08\pm1.71 \\\\
> > > 10&717.23\pm58.66&297.25\pm324.97 \\\\
> > > 18&716.35\pm57.28&406.47\pm350.39 \\\\
> > > \\hline
> > > \end{array}
> > > $$
> > >
> > > Quadruped: Run
> > > $$
> > > \begin{array}{ccc}
> > > \\hline
> > > \\mathrm{Total Steps} (1e6) & \\mathrm{Old Reward} & \\mathrm{New Reward} \\\\
> > > \\hline
> > > 2&821.62\pm54.88&783.21\pm77.03 \\\\
> > > 10&965.25\pm15.98&951.15\pm28.91 \\\\
> > > 18&951.59\pm13.02&955.61\pm25.86 \\\\
> > > \\hline
> > > \end{array}
> > > $$
> > >
> > > Point Mass: Easy
> > > $$
> > > \begin{array}{ccc}
> > > \\hline
> > > \\mathrm{Total Steps} (1e6) & \\mathrm{Old Reward} & \\mathrm{New Reward} \\\\
> > > \\hline
> > > 2 & 913.94 \pm 62.21 & 885.28 \pm 26.44 \\\\
> > > 10 & 929.75 \pm 32.82 & 902.26 \pm 41.81 \\\\
> > > 18 & 905.96 \pm 25.66 & 924.83 \pm 44.67 \\\\
> > > \\hline
> > > \end{array}
> > > $$
> > >
> > > Manipulator: Bring Ball
> > > $$
> > > \begin{array}{ccc}
> > > \\hline
> > > \\mathrm{Total Steps} (1e6) & \\mathrm{Old Reward} & \\mathrm{New Reward} \\\\
> > > \\hline
> > > 2&0.00\pm0.00&0.00\pm0.00 \\\\
> > > 10&189.31\pm378.41&1.62\pm5.13 \\\\
> > > 18&610.85\pm295.04&143.30\pm302.60 \\\\
> > > \\hline
> > > \end{array}
> > > $$
> > >
> > > Cartpole: Three Poles
> > > $$
> > > \begin{array}{ccc}
> > > \\hline
> > > \\mathrm{Total Steps} (1e6) & \\mathrm{Old Reward} & \\mathrm{New Reward} \\\\
> > > \\hline
> > > 2&219.64\pm6.07&240.66\pm42.45 \\\\
> > > 10&499.05\pm109.37&415.99\pm157.16 \\\\
> > > 18&532.93\pm227.68&558.91\pm102.08 \\\\
> > > \\hline
> > > \end{array}
> > > $$
> > >
> > > Dog: Run
> > > $$
> > > \begin{array}{ccc}
> > > \\hline
> > > \\mathrm{Total Steps} (1e6) & \\mathrm{Old Reward} & \\mathrm{New Reward} \\\\
> > > \\hline
> > > 2&27.58\pm28.74&35.84\pm67.82 \\\\
> > > 10&138.16\pm205.54&216.62\pm270.20 \\\\
> > > 18&446.78\pm416.41&486.13\pm283.23 \\\\
> > > \\hline
> > > \end{array}
> > > $$
> > >
> > > Humanoid: Run
> > > $$
> > > \begin{array}{ccc}
> > > \\hline
> > > \\mathrm{Total Steps} (1e6) & \\mathrm{Old Reward} & \\mathrm{New Reward} \\\\
> > > \\hline
> > > 2&1.46\pm0.68&25.56\pm51.51 \\\\
> > > 10& 253.92\pm77.59&284.51\pm89.45 \\\\
> > > 18&418.07\pm28.90&459.54\pm111.50 \\\\
> > > \\hline
> > > \end{array}
> > > $$
> > >
> > > Humanoid: Stand
> > > $$
> > > \begin{array}{ccc}
> > > \\hline
> > > \\mathrm{Total Steps} (1e6) & \\mathrm{Old Reward} & \\mathrm{New Reward} \\\\
> > > \\hline
> > > 2&71.63\pm110.61&204.18\pm278.80 \\\\
> > > 10&724.10\pm85.79&764.26\pm13.00 \\\\
> > > 18&749.74\pm44.65&790.56\pm25.43 \\\\
> > > \\hline
> > > \end{array}
> > > $$
> > >
> > > On tasks where the final reward dropped, this was primarily due to some fraction of seeds where the reward did not take off. The learning curve was roughly the same for seeds for where the reward did take off.
> > >
> > > 2. The reviewer is correct that imitation learning is not necessary to get standard RL methods to make progress on some two-handed manipulation tasks. Our point was that the most recent state-of-the-art results, such as on ALOHA, do tend to use imitation learning for the most challenging tasks. But even if we wanted to compare against results on Bi-DexHands which don’t use imitation learning, it is extremely challenging to implement learning in an entirely new environment in the time allotted to rebuttals. The training setup in the Bi-DexHands repository is substantially different from the setup used in the paper - the environments are based on Isaac Gym rather than MuJoCo, the learning algorithms are implemented in PyTorch rather than JAX, etc. The hyperparameters used for training are also substantially different (number of actors, number of backup steps, trajectory lengths, etc) making comparison across experiments difficult. Setting up and running these experiments properly on our infrastructure would take at least several weeks, beyond the timeframe available for rebuttals.
> > >
> > > Nevertheless, we made a good-faith attempt to get WPO running on Bi-DexHands as quickly as possible, but quickly ran into serious issues building and running Isaac Gym, as it has been deprecated in favor of Isaac Lab. While we agree that results on Bi-DexHands would improve the paper and have no doubt that we could get experiments running given a reasonable amount of time, there is simply no way that we will be able to provide results in time for the end of the rebuttal period. We hope that you understand.

---

### Official Review · Reviewer_A8Ro · 2025-03-14

**Overall Recommendation:** 4

**Summary:**

The paper introduces a novel policy gradient update using Wasserstein Gradient Flows called Wasserstein Policy Optimization. While classical policy gradient update works with stochastic policy, it does not take gradient through the action-value space, while deterministic policy gradients are able to take gradients through the action value function but are designed for deterministic policies. Using the notion of the vector flows derived from Wasserstein-2 distance and second order Fisher approximation of the KL divergence, the authors derive a policy gradient update that's best of both worlds. They derive a tractable practical algorithm for the special case of Gaussian policies.

**Claims And Evidence:**

The authors claim to use Wasserstein gradient flows to derive policy gradient updates that have best of both worlds: use stochastic policies and gradients through the action value function. They derive the gradient update that has this property. But what concerning are:

(1) Its not always simple to derive a practical algorithm for the gradient update. For a special case of policy distribution: normal, the authors derived a tractable approximate algorithm for these updates but how generalizable is this?

(2) While WPO performs robustly well, there are environments where WPO fails. It would be interesting to see why and in what situations WPO fails.

**Essential References Not Discussed:**

To my knowledge most of the related work has been referred to.

**Experimental Designs Or Analyses:**

(1) Some baselines like the one mentioned above (prior attempts to use Wasserstein Gradient Flows) should be compared against.

**Methods And Evaluation Criteria:**

The proposed method and evaluation criteria though make sense. Though the authors didn't compare against prior Wasserstein based policy gradient methods like [1]. Comparisons against these methods will justify if their gradient updates are the ideal way of using Wasserstein Gradient Flows.

[1]: Zhang, Ruiyi, et al. "Policy optimization as wasserstein gradient flows." International Conference on machine learning. PMLR, 2018.

**Other Comments Or Suggestions:**

Nil

**Other Strengths And Weaknesses:**

Nil

**Questions For Authors:**

Please address the concerns raised.

**Relation To Broader Scientific Literature:**

Policy gradient methods updates are used in a majority of RL algorithms. There have been two commonly used updates: classical and DPG. This work proposes a novel policy gradient update rule using Wasserstein gradient flows to have best of both worlds. But it is not clear if this method scales to other policy distributions (i.e. practical algorithms designed for them).

**Theoretical Claims:**

The theoretical claims made are justified and proven.

---

> ### Author Rebuttal · Authors · 2025-03-31
>
> Thank you for your helpful comments. We appreciate the overall positive evaluation of the paper, and wanted to address the two points raised in “Claims and Evidence”:
>
> 1. To go from the idealized WPO update to the practical algorithm presented in the paper, two approximations were made: the Fisher information matrix was approximated by the FIM for a diagonal Gaussian distribution, and an entropy regularization term was added.
>
> First, there is no reason that the “bare” WPO update (i.e. without the Fisher term) couldn’t be used for general distributions over actions. As noted in the paper, the variance in the update would grow as the policy becomes more deterministic, but with gradient clipping this might be manageable. Beyond that, there are many classes of distributions for which the exact Fisher information matrix is known (e.g. exponential families) and, going even beyond that, numerous practical approximations to natural gradient descent exist which could be applied here, e.g. KFAC (Grosse and Martens 2015). Ultimately, there are going to be a spectrum of approaches with the exact update at one end and the “bare” update at the other which can be explored in future work.
>
> Second, regularization of the policy update through penalties or trust region methods is a standard technique, and the approach used here could be extended to more complex distributions over action spaces so long as an appropriate way to estimate the KL divergence between action distributions exists. For instance, a sampling-based approximation could be used for generic distributions, or an upper bound or other approximation to the KL could be used in cases where a closed form solution does not exist. In practice, the exact measure of divergence between action distributions is probably not that important, as long as it is simple to compute and is well behaved.
>
> 2. Understanding the precise cause of WPO underperforming other algorithms in certain environments is challenging. Note that we only tried tuning a small number of parameters, primarily the weights of the KL regularization, while other methods have benefited from years of extensive hyperparameter tuning on many of these environments to achieve optimal performance. Note also that, while WPO is not always the top performing algorithm, it does not fail dramatically on any environments in the way that, say, DDPG does on Cartpole - Swingup Sparse, which illustrates the overall robustness of WPO. We agree that it would be quite valuable to investigate the cases where WPO underperforms in more detail, but given the large number of moving parts and possible causes, we feel this is best left to future work. Furthermore, while Figs. 3 and 6 show that WPO is competitive over the DM control suite, our main performance claims are highlighted in Figs. 4 and 5 that show that, as conjectured, WPO especially shines in domains with relatively high-dimensional action spaces.
>
> We also appreciate the suggestion to compare against Zhang et al (2018). The algorithm implemented in Zhang (2018) is a combination of a DDPG-like gradient with the reparameterization trick, and thus closely resembles SVG(0) and SAC, albeit with a slightly different regularization. The significance of WPO lies mainly in its generality, going beyond methods like SVG(0), SAC and Zhang (2018) that rely on reparameterization. Additionally, we believe that by including SAC and DDPG in the experiments already, we have good coverage of methods with the same basic form of the policy gradient update. If we investigate the effect of regularization in more detail in future work we will be sure to include an analysis of Zhang (2018). Once again, we are pleased that you liked the paper, and appreciate all of your helpful comments.

---

### Decision · Program_Chairs · 2025-05-01

**Decision:**

Accept (poster)

**Comment:**

The paper proposes a novel actor critic algorithm, Wasserstein Policy Optimization, that uses Wasserstein gradient flows to optimize stochastic policies using the gradient of the state-action value function without relying on reparameterization tricks.
However some reviewers do raise the issue of weak empirical evaluation and baselines performing below par.